# Selftok-Zero: Reinforcement Learning for Visual Generation via Discrete and Autoregressive Visual Tokens

**Bohan Wang[2][*]**  **Mingze Zhou[1,2][*][‡]**  **Zhongqi Yue[3][*]**  **Wang Lin[1]**  **Kaihang Pan[1]**  **Liyu Jia[3]**
**Wentao Hu[3]**  **Wei Zhao[2]**  **Hanwang Zhang[2][†]**

[1]Zhejiang University  [2]Huawei Central Media Technology Institute  [3]Nanyang Technological University
{mingze, linwanglw, kaihangpan}@zju.edu.cn
{liyu002, wentao002}@e.ntu.edu.sg  bohan.wang97@gmail.com
{zhaowei82, zhanghanwang}@huawei.com

## Abstract

Reinforcement learning (RL) has become an indispensable post-training step for unlocking the full potential of Large Language Models (LLMs). Its core motivation is to incentivize the model's inference trajectory via a reward model, effectively balancing the exploration–exploitation trade-off in scenarios where collecting exhaustive input–output ground-truth pairs is infeasible. This motivation naturally extends to visual generation, where perfect alignment between an image and a textual prompt is inherently ambiguous and often unattainable. However, existing visual generative models are not yet ready for RL due to the following two fundamental drawbacks that undermine the foundations of RL:

• For diffusion-based models, the actual generation trajectories of sampled images cannot be reliably rewarded, as diffusion inversion is notoriously difficult.
• For autoregressive (AR) models, we show that the widely used spatial visual tokens do not satisfy the Bellman equation and thus violate the policy improvement theorem of RL.

To this end, we propose to use Selftok (Self-consistency Tokenizer), which represents each image as a sequential 1D stream of discrete, autoregressive tokens. Together with language, we train a pure AR vision-language model (VLM) for visual generation. Impressively, without using any text-image training pairs, a simple policy gradient algorithm applied to Selftok tokens significantly boosts visual generation performance, surpassing existing models by a large margin. Implementation details are provided in the Appendix.

## 1 Introduction

Recent advances in visual generative models have been driven by large-scale training on paired text-image datasets [57, 13]. To produce high-fidelity image $\mathbf{x}$ given textual prompt $y$, these models decompose the complex image generation process into a sequence of simpler steps. In diffusion-based models [36, 15], this decomposition unfolds over continuous time-steps $t \in [0, 1]$ by the reverse diffusion process: starting from Gaussian noise $\mathbf{x}_0 \in \mathcal{N}(0, 1)$ with $t = 0$, the model iteratively denoises each $\mathbf{x}_t$ to form the trajectory $\mathbf{x}_0 \rightsquigarrow \mathbf{x}_1$, where $\mathbf{x}_1 = \mathbf{x}$. In autoregressive (AR) models [65, 74], $\mathbf{x}$ is represented as a sequence of discrete visual tokens $\mathcal{V}_K = [v_1, \ldots, v_K]$ (reasons not considering continuous visual tokens in this work included in Appendix A.1), and image generation proceeds by sequentially predicting each token given previous ones. In both cases, models

---
[*] Equal Contribution.  [†] Corresponding Author.  [‡] Research done during internship at Huawei.

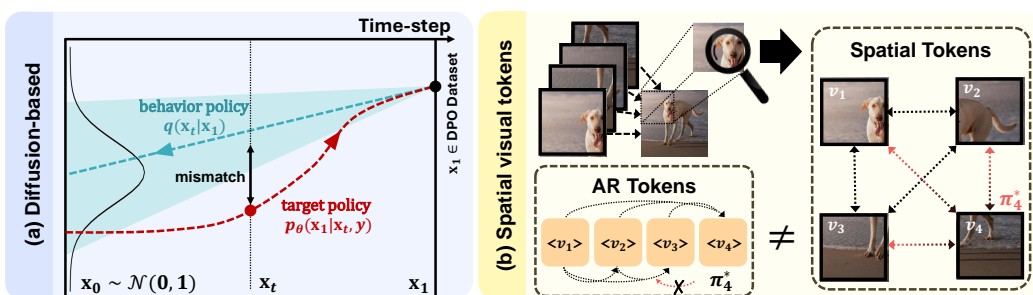

Figure 1: (a) The mismatch between the behavior policy $q(\mathbf{x}_t|\mathbf{x}_1)$ and target policy (the reverse trajectory of a diffusion model parameterized by $\theta$) in Diffusion-DPO [69]. This leads to poor action-space coverage, *e.g.*, $\mathbf{x}_t$ along the target policy trajectory is often outside the shaded 95% confidence interval of $q(\mathbf{x}_t|\mathbf{x}_1)$). (b) Due to the anti-causal links (red) for spatial tokens, learning the locally optimal policy $\pi_4^*$ for a later token (*e.g.*, $v_4$) can propagate backward and interfere with earlier tokens that were already optimized (*e.g.*, $v_1, v_2, v_3$). In contrast, AR tokens without such links do not have this issue. A more formal illustration is in Section 2.2.

are trained to follow the alignment of each paired training sample $(\mathbf{x}, y)$ over the entire generation process: given $y$, each generation step is directly supervised to match a ground-truth specified by $\mathbf{x}$, *i.e.*, its forward path in diffusion-based models or token sequence in AR models.

However, this form of supervised learning is fundamentally limited due to the one-to-many nature of text-to-image mapping. For instance, a single text prompt may correspond to infinitely many plausible images, yet the training dataset only includes a finite set of its typical looks (*e.g.*, $\mathbf{x}$ =round apple, $y$ ="an apple"). Because perfect alignment between $\mathbf{x}$ and $y$ is inherently ambiguous, models trained through supervision eventually resort to mimicking the training distribution, rather than faithfully following prompt $y$ to generate a corresponding $\mathbf{x}$. For example, we empirically observe that in early-stage training, models can still follow prompts about an atypical look, such as "a squared apple", but as training converges, their generated images eventually collapse to typical examples in the training dataset, such as round apples.

A straightforward remedy is to curate a supervised dataset that exhaustively covers all possible alignments. Yet, this approach is unsustainable.

This motivates us to train visual generative models with Reinforcement learning (RL), which offers a proven solution to this challenge, as demonstrated extensively on Large Language Models (LLMs) [66, 24]. Instead of supervising each generation step by the alignment in the training dataset, RL instead imposes a task-specific reward only after the full generation process is complete, *e.g.*, computing the CLIP-based image-prompt similarity as the reward. This shift enables the model to explore diverse generation trajectories and exploit those that yield high rewards, thereby incentivizing prompt-following visual generation. For example, given $y$ ="a square apple", a model trained with RL is incentivized to produce any image that is semantically consistent with the prompt, without being constrained by the alignment in the training dataset.

Unfortunately, existing image representations have the following limitations for visual RL:

- Diffusion-based models induce an infinite Markov Decision Process (MDP) formulation with high-dimensional, continuous state-action spaces (*i.e.*, $\mathbf{x}_t$ as a state, a denoising step as an action), which complicates optimization in RL. Recent attempt [69] explores an off-policy approach, where the key challenge is the lack of state-action trajectories available in the original Direct Preference Optimization formulation [55], due to the intractable diffusion inversion [48]. As a workaround, it samples from the forward diffusion process to approximate the behavior policy. However, as shown in Figure 1a, this introduces a large mismatch between the behavior policy (a linear forward path) and the target policy (a non-linear reverse trajectory), leading to poor action-space coverage and hence inefficient learning [63].
- Current AR models use spatial visual tokens [9, 75, 74], where images are represented as grids of patches, a convention dating back to early computer vision. However, we show that these spatial tokens lack the true AR structure, which violates the policy improvement optimality in RL [63] (Section 2.2), as illustrated in Figure 1b. First, spatial pixels (the cause) collectively form the image (the effect), and observing any part of the image during encoding induces spurious dependencies

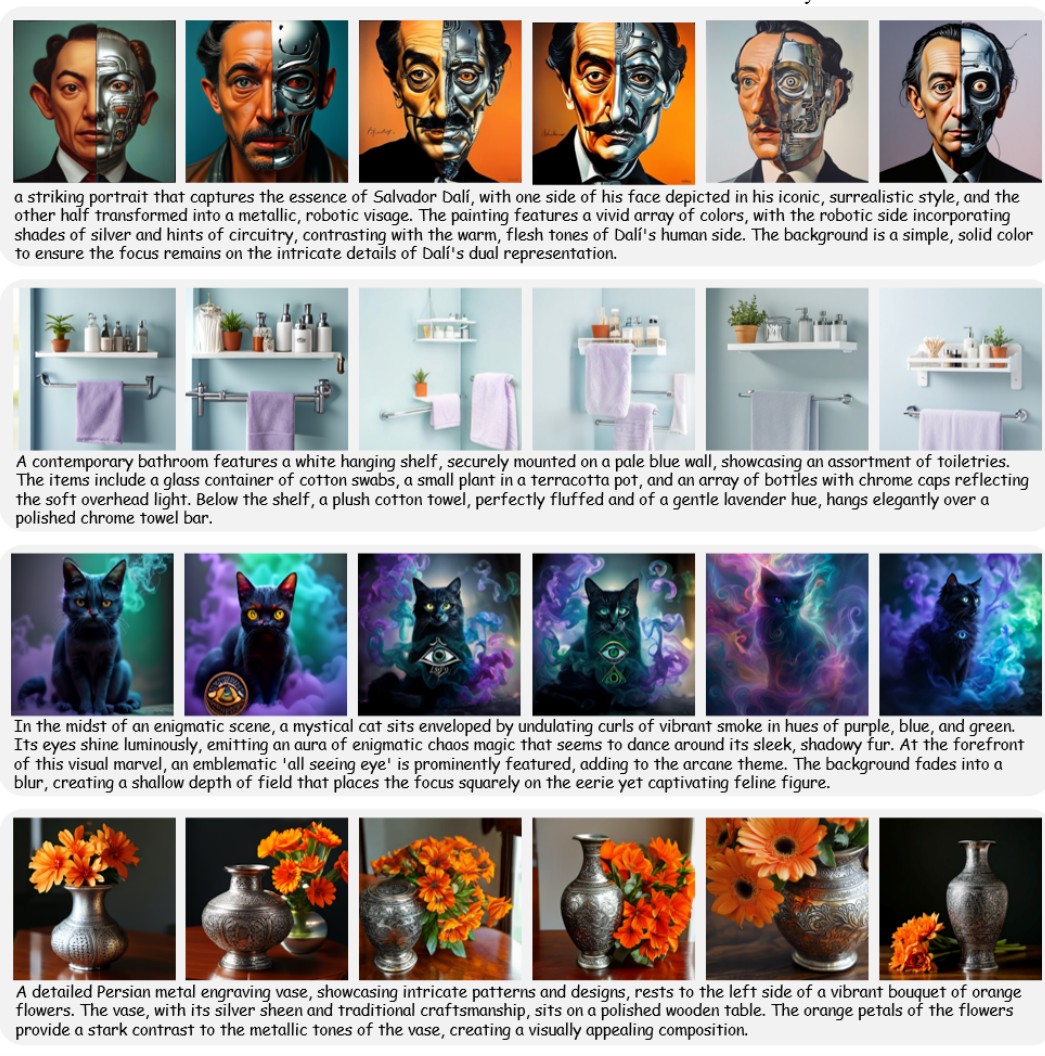

| Selftok-SFT | Selftok-Zero | Janus-Pro-7B | Janus-Pro-7B-Zero | MidJourney V6.1 | FLUX1.1 PRO |

a striking portrait that captures the essence of Salvador Dalí, with one side of his face depicted in his iconic, surrealistic style, and the other half transformed into a metallic, robotic visage. The painting features a vivid array of colors, with the robotic side incorporating shades of silver and hints of circuitry, contrasting with the warm, flesh tones of Dalí's human side. The background is a simple, solid color to ensure the focus remains on the intricate details of Dalí's dual representation.

A contemporary bathroom features a white hanging shelf, securely mounted on a pale blue wall, showcasing an assortment of toiletries. The items include a glass container of cotton swabs, a small plant in a terracotta pot, and an array of bottles with chrome caps reflecting the soft overhead light. Below the shelf, a plush cotton towel, perfectly fluffed and of a gentle lavender hue, hangs elegantly over a polished chrome towel bar.

In the midst of an enigmatic scene, a mystical cat sits enveloped by undulating curls of vibrant smoke in hues of purple, blue, and green. Its eyes shine luminously, emitting an aura of enigmatic chaos magic that seems to dance around its sleek, shadowy fur. At the forefront of this visual marvel, an emblematic 'all seeing eye' is prominently featured, adding to the arcane theme. The background fades into a blur, creating a shallow depth of field that places the focus squarely on the eerie yet captivating feline figure.

A detailed Persian metal engraving vase, showcasing intricate patterns and designs, rests to the left side of a vibrant bouquet of orange flowers. The vase, with its silver sheen and traditional craftsmanship, sits on a polished wooden table. The orange petals of the flowers provide a stark contrast to the metallic tones of the vase, creating a visually appealing composition.

Figure 2: Text-to-Image generation results by Selftok using the text prompts of DPG-Bench.

among tokens due to the collider effect [51], leading to a non-AR causal graph. Second, predicting a token at a later step (action) affects the tokens predicted in earlier steps (earlier states), so the later policy may contradict earlier policies that have already been optimized. Hence, RL applied to spatial tokens is expected to be significantly less effective than when applied to AR tokens.

In this paper, we build an AR model that supports effective RL-based post-training for visual generation. First, we completely abandon the long-standing spatial prior and introduce **Selftok: Self-consistency Tokenizer** [70], which leverages the AR nature of the reverse diffusion process to encode an image into *autoregressive* tokens corresponding to its diffusion generation trajectory (Section 2.1). Next, thanks to its AR property, Selftok produces visual tokens that satisfy the optimality condition of the policy improvement (Section 2.2). Motivated by this, we build **Seltok-Zero** (Section 3), a Selftok-based AR model post-trained with visual RL. Without using any pairwise supervision, Selftok-Zero achieves impressive image generation performances on GenEval: 92% (Table 1) and DPG-Bench: 85.57 (Table 2). Comparisons of text-to-image generation with existing VLMs are given in Figure 5.

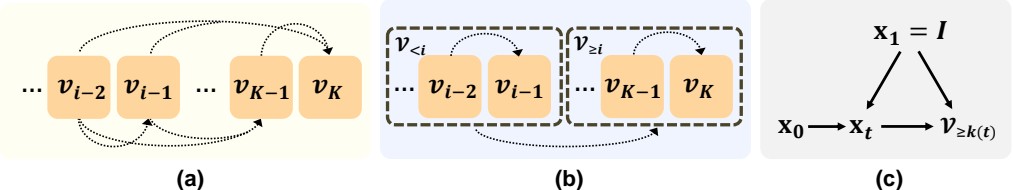

Figure 3: (a) The causal graph for AR, where each dotted direct edge represents a causation. (b) The recursion of the AR causal graph. (c) The causal graph for learning $\mathcal{V}_{\geq k(t)}$ in Eq. (5).

## 2 Problem Formulation

We begin by introducing the Selftok tokenizer [70]—which encodes images into autoregressive (AR) token sequences derived from the reverse diffusion process—and an AR model based on its visual tokens (Section 2.1). We then formulate the visual reinforcement learning (RL) problem and show why AR tokens are necessary for policy improvement in this setup (Section 2.2).

### 2.1 Selftok: Self-Consistency Tokenizer

Selftok encode an image $I$ into $K$ discrete tokens, *i.e.*, $\text{Enc}(I) = \mathcal{V}_K = [v_1, v_2, ..., v_K]$, which can be decoded to reconstruct $I$ while adhering to an autoregressive (AR) prior. We formulate the following constrained optimization:

$$\min_{\text{Enc}(I)=\mathcal{V}_K,\ \text{Dec}} \|I - \text{Dec}\left(\mathcal{V}_K\right)\|^2,$$

$$s.t.\ \ P(\mathcal{V}_K) \overset{\text{AR}}{=} P(v_1) \cdot P(v_2|v_1) \cdot \ldots \cdot P(v_K|v_1, \ldots, v_{K-1}), \tag{1}$$

where we define $\overset{\text{AR}}{=}$ as a special equality to indicate that the tokens $\mathcal{V}_K$ conform to the AR causal graph in Figure 3a, *i.e.*, each token is generated from its predecessors[1]. This definition is necessary, as the factorization is always valid for any token sequence through the chain rule of probability and does not necessarily imply an AR structure *per se*.

As with other discrete compression problems [78], solving the constrained optimization in Eq. (1) is inherently NP-hard due to the combinatorial nature of token assignment. To make this tractable, we introduce an inductive bias grounded in the reverse diffusion process, which jointly satisfies the AR constraint and the reconstruction objective. In particular, the term **"Consistency"** comes from Consistency Model [60]. Similarly, we use a diffusion model and make the decoder consistent with the image generation path, *i.e.*, reconstructing $\mathbf{x}_1 = I$ from any noisy inputs $\mathbf{x}_t$ along the path.

Specifically, we show in Figure 3b that AR structure has an equivalent recursion, enabling a divide-and-conquer approach that decomposes the challenging constraint in Eq. (1) into simpler ones:

$$P(\mathcal{V}_K) \overset{\text{AR}}{=} P(\mathcal{V}_{<i}) \cdot P(\mathcal{V}_{\geq i}|\mathcal{V}_{<i}), \tag{2}$$

where $\mathcal{V}_{<i} = [v_1, v_2, ..., v_{i-1}]$ and $\mathcal{V}_{\geq i} = [v_i, v_{i+1}, ..., v_K]$. For example, we can recursively apply Eq. (2) until it becomes a trivial learning problem $P(\mathcal{V}_{<K}) \cdot P(v_K|\mathcal{V}_{<K})$: if $\mathcal{V}_{<K}$ is provided, it is easy to encode the last token $v_K$. Interestingly, the reverse diffusion process (in ODE form) has a similar decomposition [43, 61]:

$$\frac{d\mathbf{x}_t}{dt} = \mathbf{v}_t(\mathbf{x}_t),\ \ t \in [0, 1] \quad \overset{\text{solution}}{\Longrightarrow} \quad \underbrace{\mathbf{x}_1}_{\text{destination}} = \underbrace{\mathbf{x}_t}_{\text{midway point}} + \underbrace{\int_t^1 \mathbf{v}_s(\mathbf{x}_s)ds}_{\substack{\text{path from midway} \\ \text{to destination: } \mathbf{x}_t \rightsquigarrow \mathbf{x}_1}}, \tag{3}$$

where $\mathbf{v}_t(\mathbf{x}_t)$ is the velocity field at time-step $t$ that transports the noisy midway $\mathbf{x}_t$, starting from $\mathbf{x}_0 \in \mathcal{N}(0, 1)$, towards the clean image $\mathbf{x}_1 = I$. This shows that, if the midway $\mathbf{x}_t$ is provided, the reconstruction of $\mathbf{x}_1$ starting from $\mathbf{x}_t$ is easier than directly moving from $x_0$ to $x_1$.

---

[1]This can be written mathematically as $P\left(\mathcal{V}_{<i}|do(\mathcal{V}_{\geq i})\right) = P(\mathcal{V}_{<i})\ \forall i \in \{1, \ldots, K\}$ using the do-calculus [51].

Hence, we can establish a correspondence between the two recursions by aligning the provided midway point (part 1) and what comes after it (part 2), respectively:

$$\left(\underbrace{P(\mathcal{V}_K) \Longleftrightarrow \mathbf{x}_1}_{\text{Whole}}\right) = \left(\underbrace{P(\mathcal{V}_{<i}) \Longleftrightarrow \mathbf{x}_t}_{\text{Part 1}}\right) + \left(\underbrace{P(\mathcal{V}_{\geq i}|\mathcal{V}_{<i}) \Longleftrightarrow \int_t^1 \mathbf{v}_s(\mathbf{x}_s)ds}_{\text{Part 2}}\right). \quad (4)$$

Motivated by this, we aim to compose the AR constraint into the reconstruction in Eq. (1). Specifically, we decompose the entire reconstruction (from pure noise $\mathbf{x}_0$ to $\mathbf{x}_1$) into two parts with a similar recursion: Part 1: A given $\mathbf{x}_t$, sampled from the diffusion path $q(\mathbf{x}_t|\mathbf{x}_1)$, encapsulates $\mathcal{V}_{<i}$, which is assumed to be already encoded; and Part 2: The reconstruction from $\mathbf{x}_t$ to $\mathbf{x}_1$ for learning the tokens $\mathcal{V}_{\geq i} = [v_i, v_{i+1}, \ldots, v_K]$. Now, we present the Selftok training objective for an image sample $\mathbf{x}_1 = I$:

$$\text{Selftok objective}: \min_{\substack{\text{Enc}(\mathbf{x}_1)=\mathcal{V}_K, \\ \text{Dec}}} \mathbb{E}_{t \in [0,1]} \left[ \mathbb{E}_{\mathbf{x}_t \sim q(\mathbf{x}_t|\mathbf{x}_1)} \left[ \|\mathbf{x}_1 - \text{Dec}(\mathbf{x}_t, \mathcal{V}_{\geq k(t)})\|^2 \right] \right], \quad (5)$$

where $\mathcal{V}_{\geq k(t)} = [v_{k(t)}, v_{k(t)+1}, ..., v_K]$ and $k(t)$ is a token schedule with $k(1) = K+1$ and $k(0) = 1$, which maps each continuous time-step $t$ to a discrete token index $i$ in Eq. (4). The choices of $q(\mathbf{x}_t|\mathbf{x}_1)$ and $k(t)$ are discussed in Section C. When the context is clear, we use $k(t)$ and $i$ interchangeably. We highlight that our Sefltok is indeed **non-spatial**: $\mathcal{V}_K$ discretizes the continuous velocity field of the entire image generation path, which is beyond the naïve spatial visual cues. Seltok objective in Eq. (5) optimizes the original one in Eq. (1) from three aspects: (1)Reconstruction; (2)AR Constraint by Recursive Design; (3)AR Constraint by Causal Identification. Detailed explanation are provided in Appendix A.2. Thus, the inner expectation of Eq. (5) can be rewritten as:

$$\mathbb{E}_{\mathbf{x}_0 \sim \mathcal{N}(0,1)} \left[ \|\mathbf{x}_1 - \text{Dec}\left(\sigma(t) \cdot \mathbf{x}_0 + \mu(t) \cdot \mathbf{x}_1, \mathcal{V}_{\geq k(t)}\right)\|^2 \right], \quad (6)$$

We pre-train a VLM based on the AR tokens produced by Selftok. First, we initialize the VLM from the pretrained Llama3-8B [2] model and expand its vocabulary with an additional $|\mathcal{C}| = 32,768$ Selftok visual words. As a result, the model's vocabulary integrates both textual and visual tokens into a unified embedding space. Next, the VLM is pre-trained using the standard language modeling objective on interleaved language and visual tokens. We include additional details in Appendix B.

## 2.2 Visual RL

In visual RL, we aim to fine-tune a VLM (policy) that selects the next token (action) based on the current sequence (state) to maximize a task-specific reward (*e.g.*, the consistency between the text prompt and generated image). Without loss of generality, we limit our discussion to visual tokens $[v_1, \ldots, v_K]$, as the same principle applies to language tokens. We discuss the recipe for visual RL in detail:

**1) State**: The state $s_k = [v_1, \ldots, v_k]$ is the token sequence generated by VLM at step $k \in \{1, \ldots, K\}$, and the initial state $s_0 = []$ is defined as an empty sequence.
**2) Action**: An action at step $k$ selects the next token $v_{k+1}$ from the visual codebook $\mathcal{C}$, *i.e.*, at each step, there are $|\mathcal{C}|$ possible actions to choose from.
**3) State transition**: $P(s_{k+1}|s_k, v_{k+1}) = 1$ because $s_{k+1} = [s_k, v_{k+1}]$.
**4) Reward**: Generally, the reward $r(s_k, v_{k+1})$ received at step $k+1$ depends on the previous state $s_k$ (where the reward is from) and the action as the next token $v_{k+1}$, predicted at the previous state (how the reward is obtained). With the state and action defined above, $s_{k+1} = [s_k, v_{k+1}]$, we can also write $r(s_{k+1}) = r(s_k, v_{k+1})$.
**5) Policy**: Given the current state $s_k$, the policy $\pi(v_{k+1}|s_k)$ predicts an action as the next token $v_{k+1}$. The goal of RL is to find an optimal policy $\pi$, which

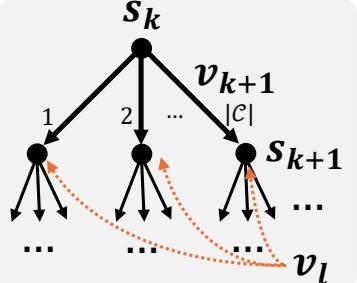

Figure 4: The recursive Bellman equation fails when a child node $v_l$ (*i.e.*, future token) anti-causally affects a parent node $v_{k+1}$.

generates a trajectory $s_0 \rightsquigarrow s_K$ that maximizes the cumulative reward (with omitted discount factor):

$$\max_{\pi} V_\pi(s_0), \quad \text{where} \quad V_\pi(s_k) = \mathbb{E}_\pi \left[ \sum_{i=k}^{K-1} r(s_i, v_{i+1}) \right]. \tag{7}$$

$V_\pi(s_k)$ is the value function, accounting for the expectation of all the possible cumulative rewards received along the trajectory $s_k \rightsquigarrow s_K$ generated by $\pi$.

We show that only AR tokens can derive the Bellman equation, which underpins the optimality of policy update that guarantees effective RL[2]. $V_\pi(s_0)$ in Eq. (7) can be rewritten as:

$$V_\pi(s_0) = \sum_{v_1 \in \mathcal{C}} \pi(v_1|s_0) \cdot [r(s_1) + V_\pi(s_1)]. \tag{8}$$

Therefore, we can recursively apply Eq. (8) and derive the Bellman equation:

$$V_\pi(s_k) = \sum_{v_{k+1} \in \mathcal{C}} \pi(v_{k+1}|s_k) \cdot [r(s_{k+1}) + V_\pi(s_{k+1})]. \tag{9}$$

Thanks to the above equation, the optimized $\pi$ in Eq. (7) can be step-by-step obtained:

$$\operatorname*{argmax}_{v_{k+1}} \pi'(v_{k+1}|s_k) \leftarrow \operatorname*{argmax}_{v_{k+1}} [r(s_{k+1}) + V_\pi(s_{k+1})]. \tag{10}$$

Although the above policy update is greedy, its optimality is guaranteed by the policy improvement theorem [63], which shows that the locally optimal action $v_{k+1}$ at step $k$ does not affect the earlier improved actions due to the AR property. Note that non-AR spatial tokens cannot satisfy the Bellman equation, and therefore cannot support the policy update that relies on it. The key reason is that Eq. (A15) cannot be derived, as the future action $v_l$, where $l > k + 1$, influences earlier actions through the anti-causal links (shown red in Figure 4). Therefore, spatial tokens are not compatible with RL.

## 3  Selftok-Zero: Selftok-based Visual RL

We now describe the implementation details for Selftok-based visual RL for visual generation, such as text-to-image and visual editing tasks, **without using any pairwise supervision**, including two reward models for evaluating the quality of the generated images and training objectives for updating the policy network.

### 3.1  Reward Model

The overall design philosophy of our reward model is to utilize visual comprehension models to evaluate the generated image in visual RL and provide feedback for optimization. For tasks such as text-to-image generation, the comprehension model should understand and evaluate the consistency between the generated image and the textual prompt. In this paper, we categorize the comprehension-based reward into two major types:

**Program-based Reward:** This type is useful for more structured tasks like object identification, counting, and spatial relationships [20], where the prompt explicitly and unambiguously states the desired generation, *e.g.*, "*3 clocks and 1 dog*", and thus we can use visual detectors [7] to evaluate the generation quality. For example, we count the clocks based on the detector's confidence, returning 1 if the count is correct and 0 otherwise. Each prompt has its own item sets to be tested, and the average of the scores for each test is used as the reward score.

**QA-based Reward:** For more complex and ambiguous prompts, it is challenging to rely solely on automated programs. To this end, we resort to more powerful visual comprehension models like InternVL [11] or GPT-4o [32], which can comprehend nuanced prompts and generate accurate answers. Specifically, inspired by [12], we first decompose the prompt to semantic tuples (*e.g.*, entity, attribute, and relation) and then generate questions (*e.g.*, "*Is the car red?*"). The MLLMs are asked to perform a VQA task for the prompt and generated image, returning a score of 0 to 1 (*e.g.*, wrong to correct) for each question. The reward is obtained by averaging the evaluation of the MLLMs

---

[2]Details of the derivation are provided in Appendix A.3

on multiple questions for a prompt. We can also fine-tune such models to obtain more task-specific reward functions.

As a preliminary study, we only validate the feasibility of the above two types. However, we believe that there should be more effective comprehension tasks as reward models for better performance, and we leave the exploration of them for future work.

## 3.2 Policy Gradient

We adopt a simplified version of GRPO [58] without importance sampling and encourage readers to explore more advanced alternatives. For each prompt, the policy network $\pi$ generates a batch of outputs $\{s^i\}_{i=1}^B$, where $B$ represents the batch size and each $s^i$ denotes the final state $[v_0, v_1, \ldots, v_K]$ of the $i$-th visual sequence. For a batch, we calculate the *total rewards* $\{r(s^i)\}_{i=1}^B$, where we slightly abuse the notation that the total reward $r(s^i) = r(s_K^i)$ as all the intermediate rewards $r(s_k^i) = 0$, $\forall k < K$. We also calculate the advantages $\{A_i\}_{i=1}^B$, where each $A_i$ measures the relative quality of output $s^i$ compared to the average reward:

$$A_i = \frac{r(s^i) - \text{mean}(\{r(s^1), r(s^2), \ldots, r(s^B)\})}{\text{std}(\{r(s^1), r(s^2), \ldots, r(s^B))}, \tag{11}$$

where $\text{mean}(\cdot)$ and $\text{std}(\cdot)$ are the mean and standard deviation of all rewards, respectively.

Then, we update the policy network parameters by the following training loss:

$$\mathcal{L} = -\frac{1}{B} \sum_{i=1}^B \left[ A_i - \lambda \mathbb{D}_{KL}(\pi || \pi_{\text{old}}) \right], \tag{12}$$

where the KL divergence $\mathbb{D}_{KL}(\pi || \pi_{\text{old}}) = \frac{\pi_{\text{old}}}{\pi} - \log \frac{\pi_{\text{old}}}{\pi} - 1$ is to maintain training stability. It measures the difference between the new policy $\pi$ and the old policy $\pi_{\text{old}}$, where the new policy $\pi$ is the up-to-date one after policy gradient; the old policy $\pi_{\text{old}}$ refers to the one used to generate the token sequences before the policy gradient update.

## 4 Related Work

Most visual generative models are trained purely on paired text–image datasets via large-scale pre-training [36, 18, 68], or additionally with supervised fine-tuning on curated high-quality data pairs [9, 19]. Recent efforts have attempted to transition from supervised learning to reinforcement learning (RL) to better align visual outputs with textual prompts. In the context of diffusion models, DPOK [17] and DDPO [5] are the first to formulate RL training frameworks, but they do not demonstrate visual generation capabilities in a fully open-vocabulary setting. Subsequent works adapt Direct Preference Optimization (DPO)[69] to diffusion-based generation, but has a mismatched behavior and target policy, leading to poor action-space coverage and inefficient learning. More recent methods based on Guided Reward Policy Optimization (GRPO)[42, 77] report promising results, but their policy networks impose restrictive Gaussian assumptions over denoising actions and operate under limited RL horizons (e.g., 10 denoising steps), which we hypothesize may hinder exploration and constrain the model's maximum potential. For AR models, existing approaches are typically built on either spatial visual tokens [74, 68, 85] or unstructured 1D token sequences [56] that lack explicit constraints to enforce causal ordering. These designs violate the policy improvement optimality in RL, which leads to significantly diminished gains from RL training. In contrast, our method builds on AR visual tokens, which enables tractable RL by defining a proper policy via softmax over a discrete, fixed-size action space. These ultimately lead to state-of-the-art performance in open-vocabulary visual generation, as demonstrated in Section 5.

## 5 Experiment

In this section, we experimentally evaluate the text-to-image generation capabilities of the Selftok-Zero, demonstrating the effectiveness of visual RL. We also provide details of the visual RL training and analyze the impact of various factors on the model performance.

Table 1: Evaluation of text-to-image generation ability on GenEval benchmark. Janus-Pro-7B†
represents the result of our evaluation. Janus-Pro-7B-Zero represents a model that has undergone the
same visual RL process as Selftok-Pre-Zero and Selftok-Zero.

| Type | Method | Single Obj. | Two Obj. | Counting | Colors | Position | Color Attr. | Overall |
|------|--------|-------------|----------|----------|--------|----------|-------------|---------|
| **Diffusion Only** | PixArt-$\alpha$ [6] | 98 | 50 | 44 | 80 | 8 | 7 | 48 |
| | SDXL [52] | 98 | 74 | 39 | 85 | 15 | 23 | 55 |
| | FLUX.1-dev [36] | 98 | 79 | 73 | 77 | 22 | 45 | 66 |
| | DALL-E 3 [59] | 96 | 87 | 47 | 83 | 43 | 45 | 67 |
| | SD3-Medium [15] | 99 | 94 | 72 | 89 | 33 | 60 | 74 |
| | CogView4-6B [3] | 99 | 86 | 66 | 79 | 48 | 58 | 73 |
| | HiDream-I1 [26] | **100** | **98** | 79 | 91 | 60 | 72 | 83 |
| **Hybrid Model** | SEED-X [19] | 97 | 58 | 26 | 80 | 19 | 14 | 49 |
| | Transfusion [82] | - | - | - | - | - | - | 63 |
| | D-DiT [40] | 97 | 80 | 54 | 76 | 32 | 50 | 65 |
| | Show-o [76] | 98 | 80 | 66 | 84 | 31 | 50 | 68 |
| | GPT-4o‡ [49] | 99 | 92 | 85 | 91 | 75 | 66 | 85 |
| **Pure dAR** | Emu3-Gen [74] | 98 | 71 | 34 | 81 | 17 | 21 | 54 |
| | TokenFlow-XL [53] | 95 | 60 | 41 | 81 | 16 | 24 | 55 |
| | ILLUME+ [31] | 99 | 88 | 62 | 84 | 42 | 53 | 72 |
| | Infinity [25] | - | 85 | - | - | 49 | 57 | 73 |
| | Janus-Pro-7B [9] | 99 | 89 | 59 | 90 | 79 | 66 | 80 |
| | Janus-Pro-7B† | 98 | 88 | 58 | 88 | 76 | 65 | 79 |
| | Janus-Pro-7B-Zero | $98_{+0}$ | $95_{+7}$ | $58_{+0}$ | $89_{+1}$ | $90_{+14}$ | $81_{+16}$ | $85_{+6}$ |
| | Selftok-Pre | 99 | 57 | 58 | 81 | 22 | 43 | 60 |
| | Sefltok-Pre-Zero | $99_{+0}$ | $94_{+37}$ | $58_{+0}$ | $89_{+8}$ | $89_{+67}$ | $73_{+30}$ | $84_{+24}$ |
| | Selftok-SFT | **100** | 79 | 66 | 91 | 45 | 62 | 74 |
| | Selftok-Zero | $99_{-1}$ | $95_{+16}$ | $\mathbf{88}_{+22}$ | $\mathbf{94}_{+3}$ | $\mathbf{96}_{+51}$ | $\mathbf{79}_{+17}$ | $\mathbf{92}_{+18}$ |

## 5.1 Implementation details

We perform visual RL (Section 3.1) on two pre-training checkpoints—Selftok-Pre trained purely on
image-text interleaved data and Selftok-SFT with additional fine-tuning on curated dataset—leading
to two final models Selftok-Pre-Zero and Selftok-Zero, respectively. We evaluate their performance
on Geneval [20] and DPG-Bench [30]. For program-based reward, we use MM-Detection [7] as the
detectors and set the threshold for detection to 0.6. For QA-based reward, we utilize InternVL [11]
and mPLUG [37] as the comprehension model. Note that we carefully deduplicate the training
prompts to ensure that there is no overlap with the test set. For the sake of reproducibility, after the
visual RL training, **we do not incorporate any test-time scaling techniques during inference.**

## 5.2 Main Results

The quantitative experimental results are summarized in Table 1 and Table 2, which evaluate the
performance of our Selftok-based approach on the GenEval and DPG-Bench benchmarks.

**Selftok-Zero achieves state-of-the-art performance in text-to-image generation.** As shown in
Table 1, Selftok-Zero obtains the highest overall score of 92 on the GenEval benchmark, surpassing
all previous models, including strong baselines such as CogView4-6B (73) and HiDream-I1 (83).
Selftok-Zero also outperforms across all major sub-tasks, *e.g.*, Colors (94) and Position (96). Sim-
ilarly, on DPG-Bench (Table 2), Selftok-Zero achieves an overall score of **85.57**, outperforming
SD3-Medium (84.08) and Janus-Pro-7B (84.19). The qualitative results are presented in Figure 5, the
images generated by Selftok-Zero exhibit high-quality alignment with the textual descriptions.

**Visual RL significantly enhances image-text consistency.** A direct comparison of Selftok-SFT
*vs* Selftok-Zero and Janus-Pro-7B† *vs* Janus-Pro-7B-Zero highlights the benefits of visual RL. On
GenEval, Selftok-Zero improves upon its supervised counterpart in nearly every metric, with notable
gains in Position (45→96) and Counting (66→88). On DPG-Bench, visual RL leads to a +3.77
increase in overall score, with improvements in Entity (from 88.15→91.78) and Relation (from
93.68→95.26). These results indicate that visual RL is effective in closing the gap between generated
images and complex textual prompts.

**Selftok is more effective than spatial tokens in visual RL.** The results in Table 1 and Table 2 show
that Selftok significantly outperforms spatial token-based methods in visual reinforcement learning
(*e.g.*, Janus-Pro-7B-Zero +6 *vs* Selftok-Zero +18 on GenEval). We illustrate the reward score changes

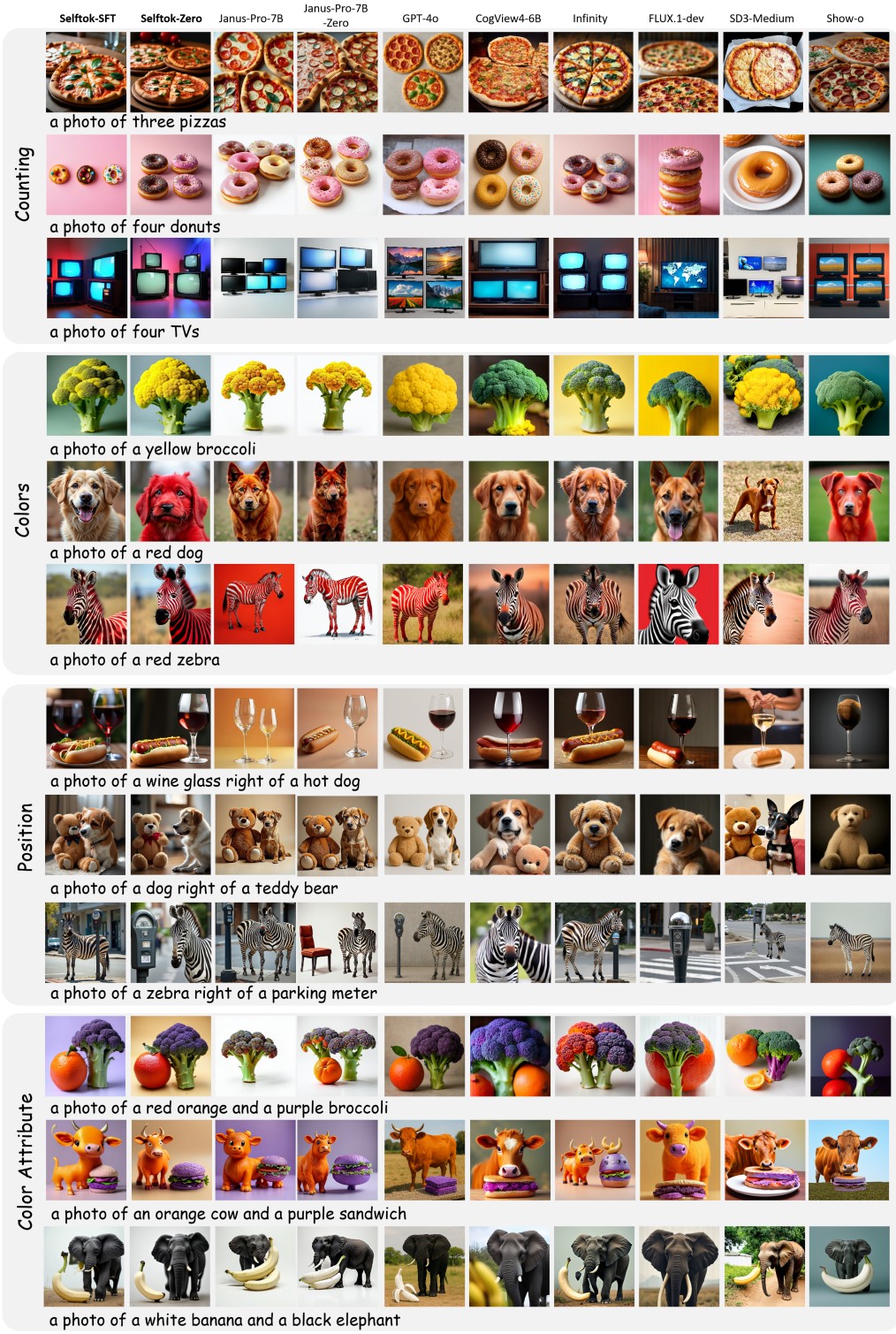

Figure 5: Qualitative experimental results of Selftok-based visual RL. Compared to existing text-to-image generation models, the images generated by Selftok demonstrate better alignment with the given prompts.

Table 2: Performances on DPG-Bench. The methods in this table are all generation-specific models except Show-o, Janus-Pro, and Selftok.

| Type | Method | Global | Entity | Attribute | Relation | Other | Overall |
|---|---|---|---|---|---|---|---|
| **Diffusion Only** | PixArt-$\alpha$ [6] | 74.97 | 79.32 | 78.60 | 82.57 | 76.96 | 71.11 |
| | SDXL [52] | 83.27 | 82.43 | 80.91 | 86.76 | 80.41 | 74.65 |
| | DALL-E 3 [59] | 90.97 | 89.61 | 88.39 | 90.58 | 89.83 | 83.50 |
| | SD3-Medium [15] | 87.90 | 91.01 | 88.83 | 80.70 | 88.68 | 84.08 |
| | FLUX.1-dev [36] | 85.80 | 86.79 | 89.98 | 90.04 | 89.90 | 83.79 |
| | CogView4-6B [3] | 83.85 | 90.35 | **91.17** | 91.14 | 87.29 | 85.13 |
| | HiDream-I1 [26] | 76.44 | 90.22 | 89.48 | 93.74 | **91.83** | **85.89** |
| **Hybrid Model** | Show-o [76] | - | - | - | - | - | 67.48 |
| **Pure dAR** | Emu3-Gen [74] | 85.21 | 86.68 | 86.84 | 90.22 | 83.15 | 80.60 |
| | Janus [75] | 82.33 | 87.38 | 87.70 | 85.46 | 86.41 | 79.68 |
| | Infinity [25] | **93.11** | - | - | 90.76 | - | 83.46 |
| | Janus-Pro-7B [9] | 86.90 | 88.90 | 89.40 | 89.32 | 89.48 | 84.19 |
| | Janus-Pro-7B† | 83.59 | 89.74 | 87.51 | 92.94 | 81.20 | 83.48 |
| | Janus-Pro-7B-Zero | 84.50$_{+0.91}$ | 90.13$_{+0.39}$ | 87.29$_{-0.22}$ | 93.44$_{+0.50}$ | 82.40$_{+1.20}$ | 84.49$_{+1.01}$ |
| | Selftok-Pre | 87.41 | 87.09 | 88.08 | 87.89 | 87.42 | 80.37 |
| | Selftok-SFT | 82.07 | 88.15 | 87.69 | 93.68 | 80.40 | 81.80 |
| | Selftok-Zero | 83.59$_{+1.52}$ | **91.78**$_{+3.63}$ | 89.04$_{+1.35}$ | **95.26**$_{+1.58}$ | 82.80$_{+2.40}$ | 85.57$_{+3.77}$ |

during visual RL evaluation on GenEval and DPG-Bench in Appendix. It is evident that although Janus-Pro-7B$^{\dagger}$ (79) outperforms Selftok-SFT (74) before visual RL, Selftok-Zero comes from behind to surpass Janus-Pro-7B-Zero (*e.g.*, +7 on Geneval), thanks to the AR properties of Selftok (see Section 2.1). These results further highlight the significant impact of the image tokenizer design on visual RL.

**Program-based reward yields more substantial gains in visual RL.** We observe that the improvements on GenEval (program-based reward) are more pronounced than on DPG-Bench (QA-based reward). While Selftok-Zero outperforms Selftok-SFT by +18 in overall score on GenEval (74→92), the improvement on DPG-Bench is slightly smaller (+3.77, 81.80→85.57). This suggests that program-based reward—enabled by structured detectors and precise matching—provides stronger and more reliable training signals during reinforcement learning, especially for attributes like object counting, color, and spatial layout.

**Qualitative Examples.** In Figure 2, we visualize the performance of Selftok-Zero on the DPG test prompt. We also compare our model with MidJourney [67] and FLUX [36], showing that Selftok-Zero performs well in both adhering to complex semantics and generating aesthetically pleasing images. However, it should be noted that the current model can only generate images at a resolution of $256 \times 256$, indicating significant potential for improvement in image detail in future work.

# 6 Conclusion

In this paper, we introduce Selftok-Zero, an autoregressive (AR) visual generative model trained with reinforcement learning (RL)), built upon Selftok's AR visual token representation. Unlike prior models based on spatial or unstructured token sequences, Selftok-Zero leverages the AR dependency among Selftok tokens to enable stable and theoretically grounded policy improvement under RL. By defining a well-structured policy over a tractable discrete action space, Selftok-Zero eliminates the need for pairwise supervision and enables efficient, end-to-end RL optimization using task-specific reward signals. Empirically, Selftok-Zero achieves strong results, significantly outperforming existing models on GenEval and DPG-Bench benchmarks. To our knowledge, this is the first work to demonstrate that RL-based post-training can substantially enhance the open-vocabulary visual generation capabilities of AR models. As future work, we aim to improve the token generation speed of Selftok-Zero by spatial-temporal compression, and extend Selftok-Zero toward high-resolution generation and physics-aware visual reasoning.

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

# A  Additional Formulation

## A.1  Why Discrete?

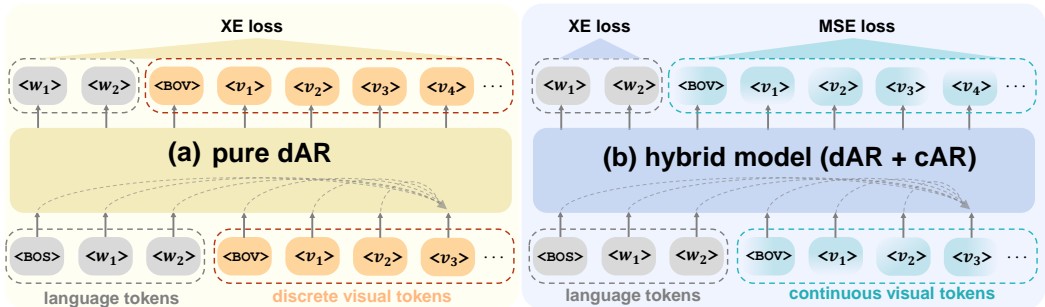

Figure A6: Comparison of (a) pure discrete autoregressive model (dAR) and (b) hybrid model that combines dAR and continuous autoregressive model (cAR). <BOS>/<BOV> indicates the start of a sentence/image. $<w_i>$/$<v_i>$ denotes the $i$-th language/visual token. Both models predict the next token given all previous ones, e.g., [<BOS>, ..., $<v_3>$] $\rightarrow<v_4>$.

We advocate the use of a pure discrete autoregressive model (dAR) (Figure A6a), rather than a hybrid approach that combines a dAR for language and a continuous autoregressive model (cAR) for images (Figure A6b) [82, 38]. The latter is widely adopted by proponents who argue that visual data should be encoded as continuous tokens to minimize the compression loss, but this is just a minor concern—there are many post-processing methods available to ensure the precision [14, 47, 64, 39]. However, using cAR (or hybrid) leads to major issues that cannot be fundamentally resolved without adopting a pure dAR:

- **cAR cannot inherit the successful infrastructure and training paradigm of LLMs.** This is the most common reason cited in existing dAR-based VLMs [65, 74, 1, 9, 35]. Yet, the following three justifications are often overlooked by the community.
- **cAR is more error-prone in next-token prediction**. While dAR functions as a sequential token classifier trained with cross-entropy (XE) loss, cAR operates as a sequential vector regressor trained with mean squared error (MSE) loss, which is less stable and harder to optimize than XE [29, 8]. Perhaps this is the key reason why most cARs abandon the causal next-token prediction and revert to bidirectional modeling, such as demasking [38, 79] or holistic reconstruction [82, 45]. Unfortunately, they undermine the core design philosophy of the decoder-only AR: the causal dependency of tokens [54].
- **cAR introduces unnecessary complexity into reinforcement learning (RL)**. It is widely known that RL is an indispensable post-training step to unleash the power of LLMs [24]. However, cAR turns the finite Markov Decision Process (MDP) formulation of dAR—with a discrete state-action space—into an infinite MDP with a continuous state-action space, thereby complicating policy optimization [46].
- **Continuous representations are less disentangled than discrete ones**. Disentanglement uncovers the modular and true generative factors of data [27, 73], which are critical for: 1) Unbiased visual comprehension, e.g., if "color" and "object" are disentangled, the model can still recognize a

*black swan* as *swan*, even if all the training examples of *swans* are *white*; and 2) Controlled generation, if such disentanglement holds, the model can generate a *black swan* without seeing one in training. Since a real-valued vector is infinitely countable, a single continuous token may theoretically entangle all the factor combinations. As a result, achieving disentanglement would require an impractically large amount of training data to cover all the combinations [44], *e.g.*, we need $\mathcal{O}(N^M)$ images, where $N$ is #values per factor and $M$ is the #factors per image. In contrast, discrete tokens, with their limited information bandwidth, serve as a strong inductive bias that encourages disentanglement [28].

## A.2   Selftok: Self-consistency Tokenizer

Here, we verify that the Seltok objective in Eq. (5) optimizes the original one in Eq. (1) from the following three aspects:

1) **Reconstruction**: When $t = 0$, Eq. (5) already includes the reconstruction objective in Eq. (1) by considering $\|I - \text{Dec}(\mathcal{V}_K)\|^2 = \|\mathbf{x}_1 - \text{Dec}(\mathbf{x}_0, \mathcal{V}_K = \mathcal{V}_{\geq k(0)=1})\|^2$, because the latter decoder only takes in a new non-informative input: the white noise $\mathbf{x}_0$.

2) **AR Constraint by Recursive Design**: Due to the correspondence between AR and diffusion recursion in Eq. (4), Eq. (5) is a recursive breakdown of Eq. (1) by time-step $t$: $\mathcal{V}_{\geq i}$ is learned from the reconstruction $\|\mathbf{x}_1 - \text{Dec}(\mathbf{x}_t, \mathcal{V}_{\geq k(t)})\|^2$ that completes the path $\mathbf{x}_t \rightsquigarrow \mathbf{x}_1$; whereas the midway point $\mathbf{x}_t$ encapsulates $\mathcal{V}_{<i}$, which is considered to be already identified by $\mathbf{x}_0 \rightsquigarrow \mathbf{x}_t$. This satisfies the probability factorization in Eq. (2) and the causal structure in Figure 3a.

3) **AR Constraint by Causal Identification**: To ensure that the learned $\mathcal{V}_K$ is indeed of AR structure, *i.e.*, the encoder *identifies the causal effect* from $\mathcal{V}_{<i}$ to $\mathcal{V}_{\geq i}$, we need to justify that Eq. (5) is an unbiased estimate of $\mathcal{V}_{\geq i}$ from $\mathbf{x}_t$ (*i.e.*, $\mathcal{V}_{<i}$) for all $t \in [0, 1]$. To this end, we show that Eq. (5) induces the causal graph in Figure 3c: Causation $\mathbf{x}_0 \rightarrow \mathbf{x}_t \leftarrow \mathbf{x}_1$ denotes that $\mathbf{x}_t$ is sampled from $q(\mathbf{x}_t|\mathbf{x}_1)$ by mixing noise $\mathbf{x}_0$ and image $\mathbf{x}_1$; causation $\mathbf{x}_t \rightarrow \mathcal{V}_{\geq k(t)} \leftarrow \mathbf{x}_1$ denotes that the tokens $\mathcal{V}_{\geq k(t)}$ are learned from $\mathbf{x}_1$ and $\mathbf{x}_t$. In this way, $\mathbf{x}_0$ serves as an *instrument variable* (IV) [51], independent of the confounder $\mathbf{x}_1$. Recall the re-parametrization: $\mathbf{x}_t = \sigma(t) \cdot \mathbf{x}_0 + \mu(t) \cdot \mathbf{x}_1$, where $\sigma(t)$ and $\mu(t)$ can be considered as time-specific constants [43]. Thus, the inner expectation of Eq. (5) can be rewritten as:

$$\underset{\mathbf{x}_0 \sim \mathcal{N}(0,1)}{\mathbb{E}} \left[ \|\mathbf{x}_1 - \text{Dec}\left(\sigma(t) \cdot \mathbf{x}_0 + \mu(t) \cdot \mathbf{x}_1, \mathcal{V}_{\geq k(t)}\right)\|^2 \right], \tag{A13}$$

which implies that $\mathcal{V}_{\geq k(t)}$ can be directly estimated from the IV $\mathbf{x}_0$, ensuring that $\mathcal{V}_{\geq k(t)}$ learned from $\mathbf{x}_t$ is unbiased, even in the presence of the confounder $\mathbf{x}_1$.

## A.3   RL theoretical derivation

We now show that only AR tokens can derive the Bellman equation, which underpins the optimality of policy update that guarantees effective RL. We start by rewriting our goal $V_\pi(s_0)$ in Eq. (7):

$$V_\pi(s_0) = \underset{[v_1 \sim \pi(\cdot|s_0), v_2 \sim \pi(\cdot|s_1), \dots, v_K \sim \pi(\cdot|s_{K-1})]}{\mathbb{E}} \left[ r(s_0, v_1) + r(s_1, v_2) + \dots + r(s_{K-1}, v_K) \right]$$

$$\tag{A14}$$

$$= \underset{v_1 \sim \pi(\cdot|s_0)}{\mathbb{E}} r(s_0, v_1) + \underset{v_1 \sim \pi(\cdot|s_0)}{\mathbb{E}} \underbrace{\underset{[v_2 \sim \pi(\cdot|s_1), \dots, v_K \sim \pi(\cdot|s_{K-1})]}{\mathbb{E}} \left[ r(s_1, v_2) + \dots + r(s_{K-1}, v_K) \right]}_{V_\pi(s_1)}$$

$$\tag{A15}$$

$$= \sum_{v_1 \in \mathcal{C}} \pi(v_1|s_0) \cdot \left[ r(s_1) + V_\pi(s_1) \right]. \tag{A16}$$

Eq. (A14) holds because the transition probability $P(s_{k+1}|s_k, v_{k+1}) = 1$. As shown in Figure 4, Eq. (A15) holds because of the causal dependency of AR, where the choice of action $v_{k+1}$ only depends on $s_k$ and does not affect the former action $v_k$ that has already been chosen. Therefore, we can recursively apply Eq. (A16) and derive the Bellman equation:

$$V_\pi(s_k) = \sum_{v_{k+1} \in \mathcal{C}} \pi(v_{k+1}|s_k) \cdot \left[ r(s_{k+1}) + V_\pi(s_{k+1}) \right]. \tag{A17}$$

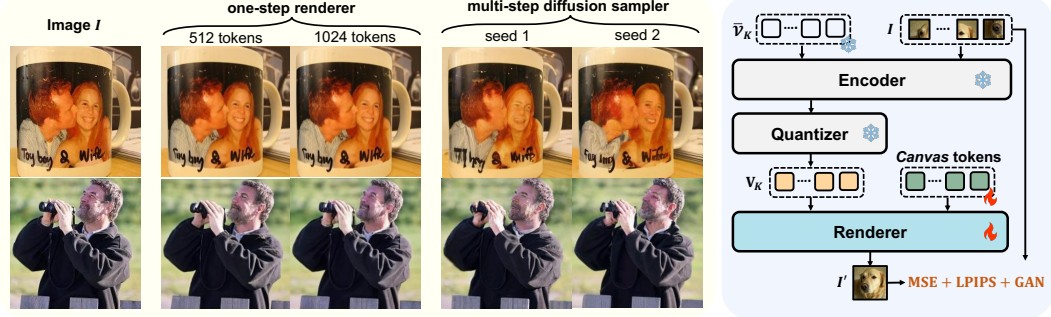

Figure B7: Left: (a) Reconstructions with one-step renderer (512 or 1024 tokens) and multi-step diffusion sampler (512 tokens, two seeds); Right: (b) Renderer architecture diagram.

# B  Additional Implementation Details

## B.1  Dataset

We conducted experiments on GenEval [20] and DPG-Bench [30]. GenEval is an object-focused framework to evaluate compositional image properties such as object co-occurrence, position, count, and color, which is under MIT License. DPG-Bench is a benchmark used to evaluate the ability of models to follow complex prompts, which is under Apache License. It contains diverse and complex prompts and constructs QA pairs based on these prompts. These QA pairs are answered using a MLM and then the final score is calculated.

## B.2  Selftok

For encoder, We use a dual-stream transformer backbone like MMDiT [15], which consists of an image stream (blue modules) and a token stream (yellow modules). Each stream has its own parameters, specialized for processing patch-based image embeddings and AR-based token embeddings. The backbone consists of $N$ blocks with identical architecture. For quantizer, we use one based on cosine similarity, which is updated through an exponential moving average instead of gradient descent. For decoder, we use a diffusion model initialized from SD3 [15]. It is a dual-stream transformer MMDiT architecture, where the input to the original language token stream is replaced with the quantized embeddings. To remove the original language influence and better adapt to Selftok tokens, the weights of our token stream are trained from scratch.

After training, we can apply a standard multi-step diffusion sampler [15, 56] to decode our tokens $\mathcal{V}_K$ into a reconstructed image. However, this process is slow as it requires multiple sequential forward passes. To accelerate this, we build a renderer $R(\mathcal{V}_K)$ that reconstructs $I$ in a single forward pass. We initialize $R$ with the decoder weights. To remove its dependency on $\mathbf{x}_t$, we replace it with a sequence of learnable "canvas" token embeddings as shown in Figure B7 (b), which becomes part of the model parameters of $R$. Then with the learned token embeddings $\mathbf{V}_K = \text{Enc}(I)$ frozen, we optimize $R$ jointly with an MSE loss for pixel-level reconstruction, LPIPS [81] and GAN [22] loss for perceptual quality, as including the latter two resolves the well-known blurry reconstruction issue when training a decoder with the MSE loss alone [34, 16]:

$$\min_{R(\mathbf{V}_K)=I'} \max_D \left[ \underbrace{\|I - I'\|^2}_{\text{MSE loss}} + \underbrace{\lambda_1 \text{LPIPS}(I, I')}_{\text{perceptual loss}} + \underbrace{\lambda_2 \left(\log D(I) + \log(1 - D(I'))\right)}_{\text{GAN loss}} \right], \quad \text{(B18)}$$

where $\lambda_1, \lambda_2$ are loss weights, $D$ is the discriminator of the GAN. To improve training stability, we set $\lambda_1 = 0.1, \lambda_2 = 0$ for the first 30k training iterations and $\lambda_1 = 0.5, \lambda_2 = 0.5$ afterwards. As shown in Figure B7 (a), besides the improved visual perception, the one-step renderer brings two benefits: 1) it significantly reduces the image generation time, and 2) it eliminates the randomness introduced by the random seed in diffusion-based generation (see Figure B7 (a)). We train the tokenizer on 32 Ascend 910B for 96 hours.

Please refer to [71] for the rest of details.

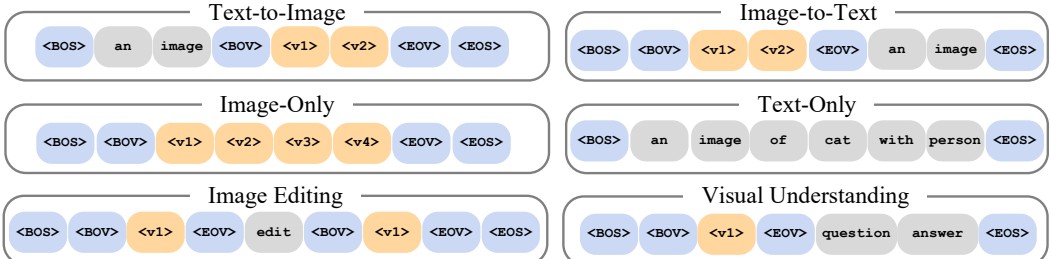

Figure B8: Illustration of the proposed data format for cross-modality and cross-task pre-training.

## B.3 Pre-training

We initialize the VLM from the pretrained Llama3-8B [2] model, which is under META LLAMA 3 COMMUNITY LICENSE, and expand its vocabulary with an additional 32,768 Selftok visual words. As a result, the model's vocabulary integrates both textual and visual tokens into a unified embedding space. The VLM is trained using the standard language modeling objective, which aims to maximize the log-likelihood of multimodal token sequences in an AR fashion:

$$P(\mathcal{Y}) = \sum_{i=1}^{|\mathcal{Y}|} \log P_\theta(y_i|\mathcal{Y}_{<i}),$$

where the sequence $\mathcal{Y}$ may consist of interleaved language and visual tokens, and thus $y_i \in \mathcal{Y}$ denotes either a language token $\langle w_i \rangle$ or a visual token $\langle v_i \rangle$. Since both text and image content are represented as discrete token IDs, the prediction head is shared and supervised at each position using a cross-entropy loss. The training consists of the following two stages:

**Stage1: Cross-modality Pre-training.** In this stage, we aim to learn the correspondence between visual tokens and language tokens, thereby facilitating the transition of the pre-trained Llama3 model from LLM to VLM. To achieve this, we introduce four data formats designed to address the challenges of cross-modality alignment. Each format helps the model process and integrate vision and language inputs for coherent multimodal understanding and generation. The *Text-to-Image* format aligns caption with visual data, enabling image generation from textual descriptions. Conversely, the *Image-to-Text* format facilitates understanding tasks by associating visual data with textual descriptions. To address potential misalignments that can occur during text-to-image tasks, the *Image-Only* format is introduced, allowing the model to learn visual structure independently. Finally, the *Text-Only* data ensures the preservation of the model's linguistic capabilities, maintaining its ability to process and generate text. These formats and their functions are summarized in Figure B8, with special tokens such as [BOS] and [EOS] marking the sequence boundaries, and [BOV] and [EOV] indicating the start and end of visual data. The training data is comprised of 530 million high-quality image-text pairs and text sequences and we train the model on 400 Ascend 910B for 120 hours.

**Stage2: Cross-task Pre-training.** In this stage, we perform cross-task pre-training to enable the model to learn human instructions across various tasks. This is accomplished through supervised fine-tuning (SFT) on datasets from three distinct tasks: 1) text-to-image generation, 2) image editing, and 3) image understanding. The instruction format follows the structure "USER: <Instructions> ASSISTANT: <Answers>", where only the content of <Answer> contributes to the loss function, optimizing the model's ability to provide accurate responses. In Stage2, we fine-tune the model using 8 Ascend 910B for 8 hours.

We denote the VLM after stage 1 and 2 as Selftok-Pre and Selftok-SFT, respectively.

## B.4 Visual RL

We adopt GRPO [58] setting the coefficient of KL divergence $\beta$ to 0.04 and the size of the group to 24 with synchronous sampling and update frequency. During training, the model is optimized using the AdamW optimizer with a learning rate of 1.5e-6. We train the model on 64 Nvidia A800 GPUs for 96 hours.

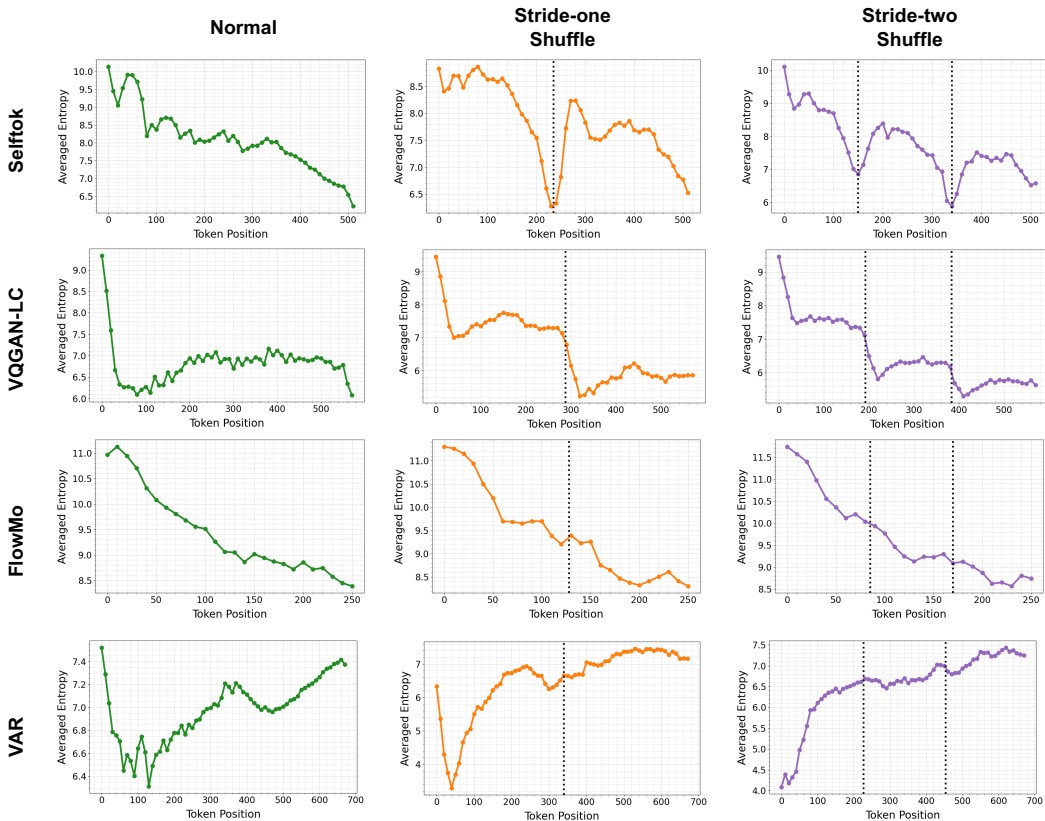

Figure C9: Plots of the next-token prediction entropy versus token position for our Selftok, 2D spatial tokens (VQGAN-LC [83]), 1D tokens (FlowMo [56]), multi-scale 2D tokens (VAR [68]), using the original or shuffled sequences. Only Selftok exhibits a segmented decreasing trend that aligns with the three sequence orders. Although VQGAN-LC also displays a segmented trend, each segment is not decreasing. Conversely, while FlowMo shows a decreasing trend, it is not segmented under the shuffled orders.

# C   Additional Results

**Semantic interpretability**. We find that tokens corresponding to smaller time-steps tend to capture the overall background, color tone or composition of the image, those at middle ones tend to capture object shapes and those at larger ones tend to capture fine-grained details and textures. This is because the diffusion process itself is tightly linked with visual semantics [1,2,3], and Selftok simply encode the process as tokens, as shown in Figure C10

**AR structure**. We empirically verify that the structure of Selftok is AR by plotting the token prediction entropy curves *w.r.t.* token positions under three generation orders using a dAR model (Llama 3.1). Besides the normal sequential order $[v_1, v_2, v_3, ...]$, we use another two orders: 1) stride-one shuffle, which is a concatenation of subsequence $[v_1, v_3, ...]$ followed by subsequence $[v_2, v_4, v_6, ...]$, and 2) stride-two shuffle, which is a concatenation of subsequence $[v_1, v_4, v_7, ...]$, $[v_2, v_5, v_8, ...]$, and $[v_3, v_6, v_9, ...]$. The design principle of these orders is simple: an ordered subsequence of an AR sequence is still AR. As entropy measures the uncertainty in token prediction, if the sequence is AR, the entropy trend is generally decreasing. Therefore, if the token sequence is AR, the two shuffled orders should demonstrate a segmented decreasing curve. As shown in Figure C9, we can see that only Selftok demonstrates such a segmented decreasing trend corresponding to the three sequence orders.

**Time sampler**. For time sampler, besides the simple uniform sampling, SD3 [15] introduces the logit-normal time-step sampler by assigning higher probability density to mid-range time-steps ($t \approx 0.5$). We compared the reconstruction performance when using uniform and logit-normal sampling in Table C3, which shows that the simple uniform sampling performs the best for Selftok.

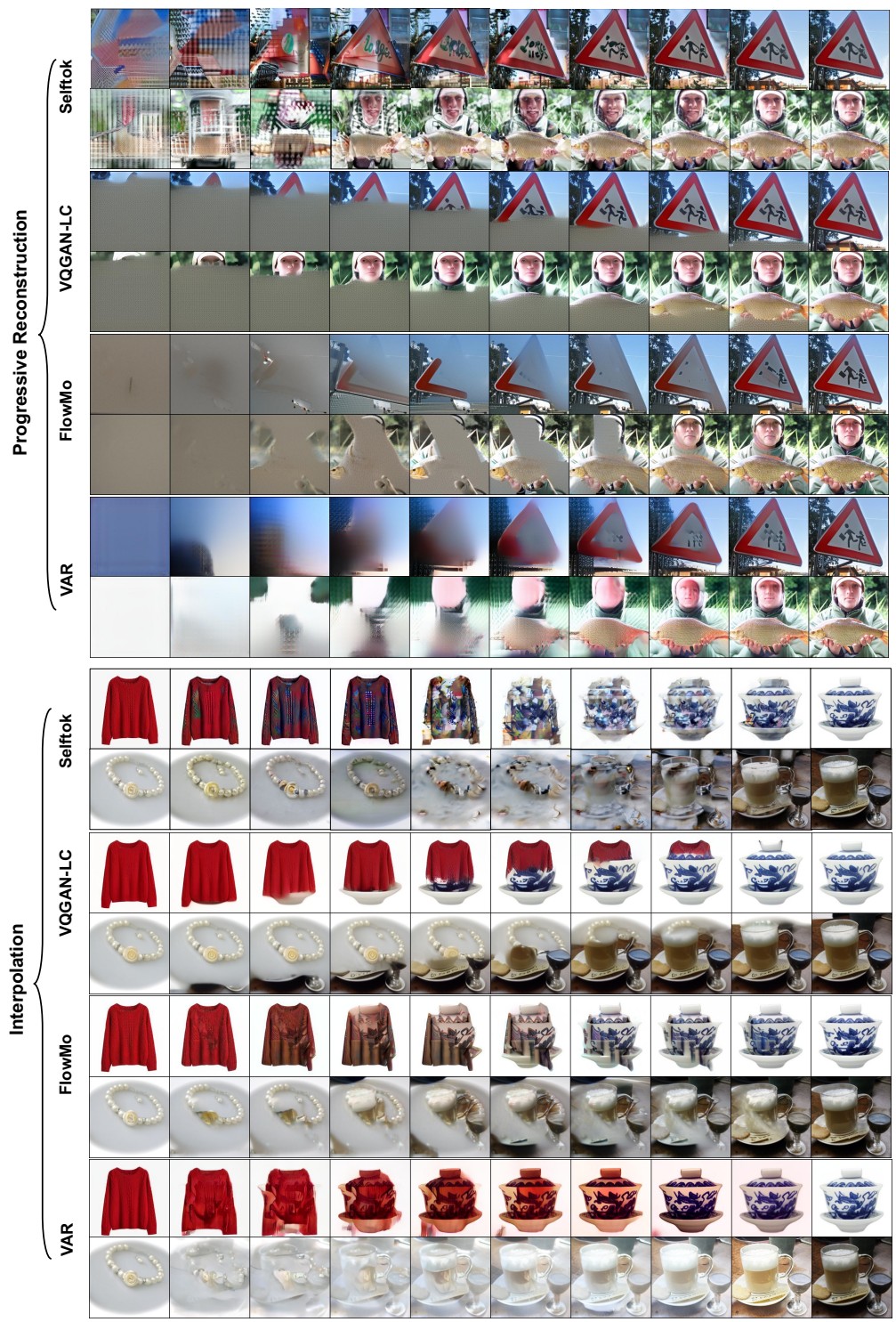

Figure C10: Progressive reconstruction (left to right): Reconstructions by progressively masking out a shorter sequence of tokens before inputting to the decoder. Interpolation (left to right): Reconstructions by gradually replacing tokens of the left image with those of the right one. All methods except Selftok exhibit strong spatial characteristics (*i.e.*, tokens⇔patches).

Table C3: Ablation on time sampler and token schedules. 'sampl.' and 'sched.' denote 'sampler' and 'schedule'.

| Time sampl. | Token sched. | PSNR↑ | SSIM↑ | LPIPS↓ |
|---|---|---|---|---|
| uniform | custom | **21.86** | **0.600** | **0.150** |
| uniform | uniform | 21.10 | 0.564 | 0.177 |
| uniform | logit-normal | 20.78 | 0.555 | 0.180 |
| logit-normal | custom | 20.98 | 0.561 | 0.170 |
| logit-normal | uniform | 19.89 | 0.498 | 0.205 |
| logit-normal | logit-normal | 20.08 | 0.513 | 0.196 |

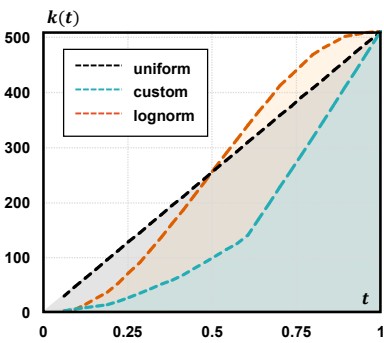

Figure C11: Token schedule $k(t)$. "lognorm" denotes logit-normal.

Table C4: Reconstruction performance of different tokenizers on $256 \times 256$-resolution ImageNet 50k validation set. [†] Results from the original paper.

| Tokenizer | Type | #Token | #Code | rFID↓ | PSNR↑ | SSIM↑ | LPIPS↓ |
|---|---|---|---|---|---|---|---|
| LlamaGen [62] | 2D | $16 \times 16$ | $2^{14}$ | 2.19 | 20.67 | 0.589 | 0.132 |
| Cosmos [1] | 2D | $32 \times 32$ | $\approx 2^{16}$ | 0.87 | 24.82 | 0.763 | 0.070 |
| VAR [68] | 2D | 680 | $2^{12}$ | 0.99 | 22.12 | 0.624 | 0.109 |
| TiTok-S-128 [80] | 1D | 128 | $2^{12}$ | 1.71 | 17.52 | 0.437 | 0.210 |
| FlexTok [4] | 1D | 256 | $64,000$ | 1.45 | 18.53 | 0.465 | 0.222 |
| FlowMo-Hi[†] [56] | 1D | 1,024 | $2^{14}$ | 0.56 | 24.93 | 0.785 | 0.073 |
| Selftok (Ours) | 1D | 1,024 | $2^{15}$ | **0.54** | **26.30** | **0.805** | **0.063** |

**Token schedule $k(t)$.** Recall that the AR constraint in Eq. (1) requires that every token must conform to the decomposition $P(\mathcal{V}_K) \overset{\text{AR}}{=} P(\mathcal{V}_{<i}) \cdot P(\mathcal{V}_{\geq i}|\mathcal{V}_{<i}), \forall i \in [1, K+1]$. We achieve this decomposition by diffusion time-steps, thanks to the recursive nature of the reverse diffusion process in Eq. (4), denoted as $\mathcal{V}_{\geq i} \Leftrightarrow \mathbf{x}_t \rightsquigarrow \mathbf{x}_1$ and $\mathcal{V}_{<i} \Leftrightarrow \mathbf{x}_0 \rightsquigarrow \mathbf{x}_t$. That is to say, the second-half tokens $\mathcal{V}_{\geq i}$ can be learned recursively by the diffusion decoder, conditioned on $\mathbf{x}_t$, which represents the already identified first-half tokens $\mathcal{V}_{<i}$. As we uniformly sample $t \in [0, 1]$ in training, the best token schedule should be a uniform assignment $k^*(t) = \lceil t \times K \rceil + 1$ to ensure that every token is involved in the recursive diffusion time-step. To better understand this, we provide three failure cases: **1)** If we allocate all the tokens to $\mathcal{V}_{\geq 1}$, *i.e.*, $k(t) = 1, \forall t \in [0, 1)$, this corresponds to a trivial decomposition $P(\mathcal{V}_{<1} = []) \cdot P(\mathcal{V}_K|\mathcal{V}_{<1} = [])$, $\mathcal{V}_K \Leftrightarrow \mathbf{x}_0 \rightsquigarrow \mathbf{x}_1$, and $[] \Leftrightarrow \mathbf{x}_0$, where we always input the full $\mathcal{V}_K$ to the decoder. So, $\mathcal{V}_K$ loses all the AR property. This case reduces to the FlowMo approach [56]. **2)** If we always allocate all tokens to $\mathcal{V}_{<1}$, *i.e.*, $k(t) = K + 1, \forall t \in (0, 1]$, this corresponds to another trivial decomposition $P(\mathcal{V}_K) \cdot P([]|\mathcal{V}_K)$, $[] \Leftrightarrow \mathbf{x}_0 \rightsquigarrow \mathbf{x}_1$, and $\mathcal{V}_K \Leftrightarrow \mathbf{x}_0$, where we always send an empty sequence to the decoder. This case reduces to the unconditional diffusion generation without learning $\mathcal{V}_K$ at all. **3)** Consider a non-extreme case where $k(t)$ is not uniformly aligned with $t$, *e.g.*, $k(t = 0.8) = \lceil 0.2 \times K \rceil$, we disrespect the decomposition because the majority of tokens $\mathcal{V}_{\geq \lceil 0.2 \times K \rceil}$ corresponds to dense time-steps in the short interval $t \in [0.8, 1]$, while the rest ones in $\mathcal{V}_{< \lceil 0.2 \times K \rceil}$ corresponds to sparse time-steps, violating the balanced recursive correspondence in Eq. (4). We explored three different choices for $k(t)$: 1) the uniform one with $k(t) = \lceil t \times K \rceil + 1$; 2) a custom schedule that allocates few tokens to small $t$; and 3) a logit-normal schedule that allocates few tokens to both small and large $t$. We plot $k(t)$ in Figure C11 and compare the performance of the models trained with each schedule in Table C3. However, in practice, we empirically observe a better reconstruction quality by designing a schedule $k(t)$ that allocates fewer tokens to smaller $t$, *i.e.*, $k(t) < k^*(t)$ for $t < 0.5$. This aligns with the well-known trait of diffusion models: the early path $\mathbf{x}_0 \rightsquigarrow \mathbf{x}_t$ for a small $t$ has minimal impact on the reconstruction $\mathbf{x}_t \rightsquigarrow \mathbf{x}_1$, which can be omitted [72, 50].

**Tokenizer metrics.** Encoding and decoding a single image with the Selftok tokenizer requires only 0.86 s and incurs a computational cost of 2.59 TFLOPs, underscoring the efficiency of our approach. Quantitative comparisons with other tokenizers are provided in Table C4; our tokenizer achieves

(a) GenEval scores of different methods.

| Methods | GenEval Score |
| --- | --- |
| SDXL | 53.8 |
| SDXL + Diffusion-DPO | 56.3 (+2.5) |
| Selftok-SFT | 74 |
| Selftok-Zero | 92 (+18) |

(b) DPG scores of different models. *Model trained only with the program-based reward.

| Model | DPG Score |
| --- | --- |
| Selftok-SFT | 81.80 |
| Selftok-P* | 82.43 |
| Selftok-Zero | 85.57 |

(c) Geneval scores of models trained with different KL-divergence coefficients.

| KL coefficient | Geneval Score |
| --- | --- |
| 0 | – |
| 0.05 | 92 |
| 0.1 | 87 |

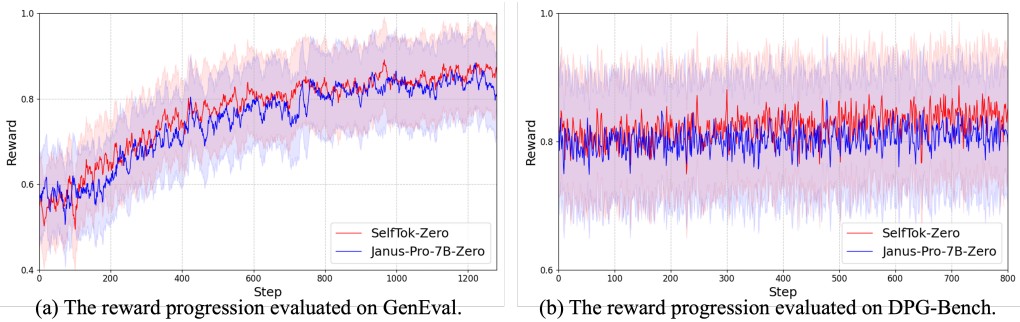

(a) The reward progression evaluated on GenEval.  (b) The reward progression evaluated on DPG-Bench.

Figure C12: Comparison of reward progression over steps for Selftok (Selftok-Zero) and spatial tokens (Janus-Pro-7B-Zero) on GenEval and DPG-Bench.

state-of-the-art performance, attaining the best results to date on rFID (0.54), PSNR (26.30), SSIM (0.805), and LPIPS (0.063).

**Selftok is more effective than spatial tokens in visual RL.** Figure C12 illustrates the reward score changes during visual RL evaluation on GenEval and DPG-Bench. It is evident that although Janus-Pro-7B[†] (79) outperforms Selftok-SFT (74) before visual RL, Selftok-Zero comes from behind to surpass Janus-Pro-7B-Zero (*e.g.*, +7 on Geneval), thanks to the AR properties of Selftok. These results further highlight the significant impact of the image tokenizer design on visual RL.

**Ablation Results**. We conducted three categories of ablation studies: 1) **Online vs. offline policy.** Using SDXL [52] with Diffusion-DPO [69] as the offline baseline, we observe in Table C5a that Diffusion-DPO underperforms our method, likely because the sample trajectories are misaligned with the model's optimization trajectories. 2) **Reward function comparison.** We compare training with only the program-based reward against training with both types of rewards. As shown in Table C5b, combining the two rewards provides a more comprehensive learning signal and yields superior performance. 3) **KL-divergence ablation.** We examine the effect of the KL divergence (Table C5c): removing the KL term leads to highly unstable training, whereas increasing the KL coefficient slows convergence. Accordingly, we set the KL coefficient to 0.05 in our experiments.

**Hallucination in Text-to-Image Generation.** One of the challenges in text-to-image generation is the "hallucination" issue, where a Vision-Language Model (VLM) tends to generate images that closely follow the training data distribution rather than genuinely reason about the text prompt. This can lead to the model failing to generate certain objects or scenes that are less common or not well-represented in the training set. In Figure C13, we provide examples where the Selftok-SFT model fails to generate certain objects due to the rarity of these combinations in the training data. However, after applying visual RL (Selftok-Zero), the model is able to generate these previously missing combinations, showing a significant improvement in handling rare or complex prompts. The ability of Selftok-Zero to generate these images after the visual RL phase highlights how reinforcement learning can effectively overcome the hallucination problem, improving the model's generalization and reasoning capabilities beyond the initial supervised training.

**Visual RL for Image Editing.** To further unlock the potential of our model, we also incorporate Visual RL into the image editing task, where we utilize a Vision-Language Model (VLM)—specifically, InternVL2.5-78B [10]—as the reward model. This model evaluates whether the generated image strictly follows the instructions and accurately modifies the source image. Inspired by the work of [23, 18], we ask the reward model to return a score between 0 and 5, with 5 indicating the highest level of adherence to the instructions. In a few hundred steps, our Selftok-Zero model shows a

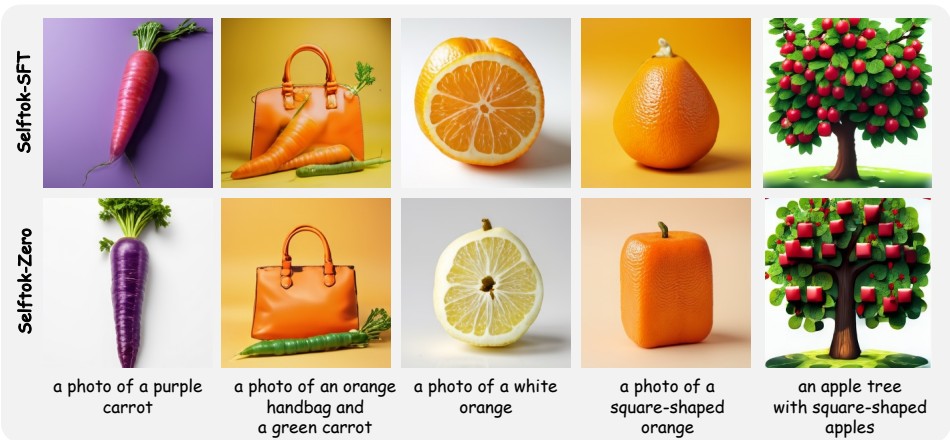

Figure C13: More examples of failures in the Selftok-SFT model due to distributional biases in the training data during vision-language supervised training.

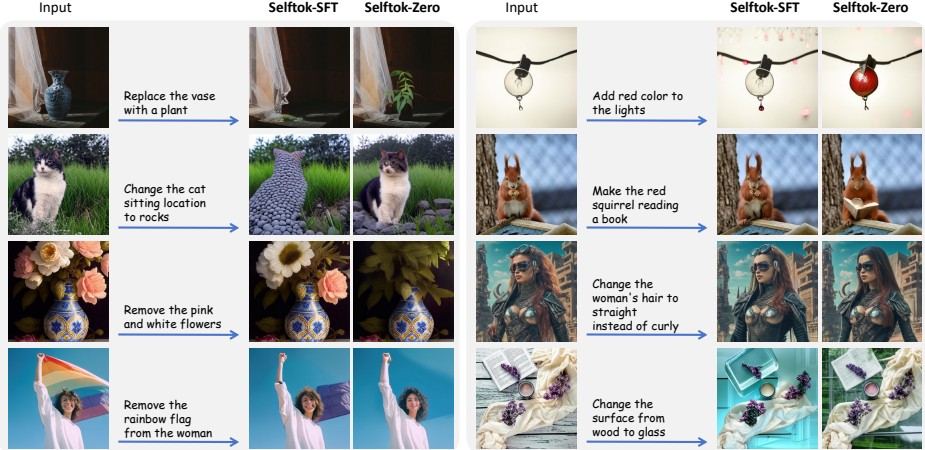

Figure C14: Qualitative experimental results of Selftok-based visual RL on image editing. Compared to the Selftok-SFT, the images generated by Selftok-Zero demonstrate better alignment with the given instructions and better visual fidelity.

significant improvement over the Selftok-SFT model. As shown in Figure C14, our model can correctly correspond to the instructions and generate appropriate edited images.

Unlike text-to-image generation, image editing involves more nuanced transformations, making it significantly more challenging to evaluate automatically. The complexity arises from the need to assess both the fidelity of the edits to the original image and the accuracy of the applied changes according to the given instructions. Therefore, to provide a more general and accurate reward for image editing tasks, we plan to explore more sophisticated reward models that can handle the intricacies of image modification. Additionally, we aim to develop refined evaluation principles that can better capture the subtlety and precision required in image editing. This will be a key focus in our future work, where we hope to improve the reliability of automated assessments and provide more meaningful feedback.

## D  Limitations

The primary limitation does not lie in Selftok itself, but rather in the significantly slower token generation speed of LLMs compared to diffusion models. For instance, when using 512 tokens per frame, generating a one-minute video clip at 24 fps would require generating $512 \times 24 \times 60 = 737,280$ tokens—posing a substantial throughput challenge. Fortunately, we are optimistic that this issue will be mitigated by introducing spatial-temporal compression, in conjunction with the rapid progress in real-time massive token generation within the LLM community [21]. Another limitation of this work stems from the restricted model scale. Due to limited capacity, we have not yet demonstrated Selftok's

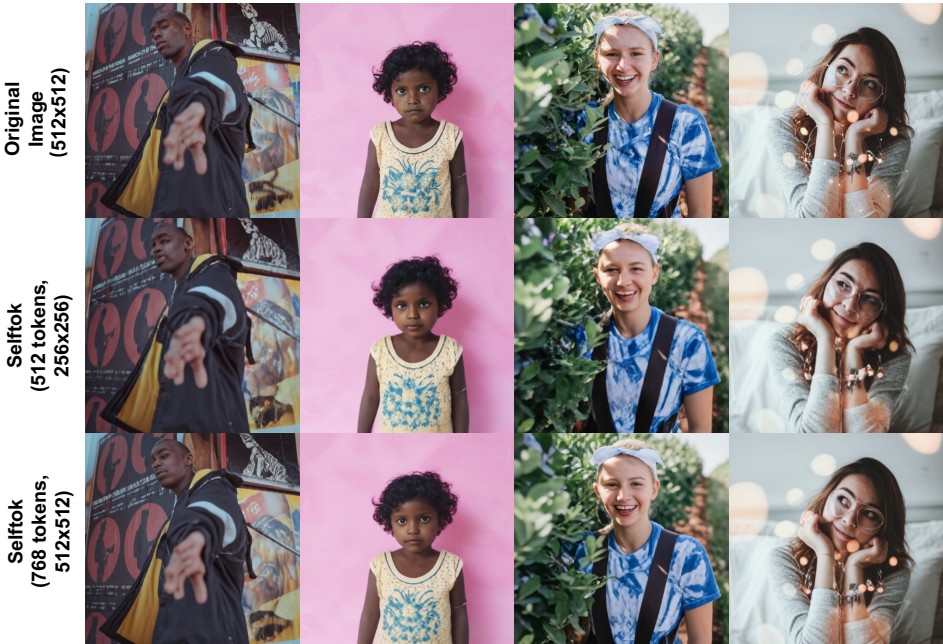

Figure C15: Qualitative results of $512 \times 512$ resolutions.

ability to transfer visual knowledge to language and realize multimodal emergent capabilities. If resources permit, we plan to investigate the scaling laws of multimodal training with Selftok, aiming to validate its potential for cross-modal synergy. Next, we highlight our two ongoing works for Selftok:

**Multi-resolution Selftok**. The current resolution of Selftok is limited to $256 \times 256$, which constrains the quality of visual generation. Our design follows an incremental principle: higher-resolution images are supported by increasing the number of tokens, while reusing the tokens extracted from their lower-resolution counterparts. This enables efficient scalability, allowing higher-resolution data to leverage a dAR model pre-trained on lower-resolution inputs. This approach is particularly appealing, as it parallels the practice in LLM training, where longer document training benefits from prior training on shorter texts. Figure C15 presents our preliminary results, which will be included in the future work.

**Physics-aware Post-training** Inspired by the impressive performance gains of visual RL by using the program-based reward, our next step is to incorporate physical laws into Selftok-based video generation. For example, we can track the trajectories of moving objects and evaluate whether they conform to fundamental motion principles. This direction has great potential in addressing the ever-lasting criticisms that large visual models struggle to learn a true world model [84, 33]. In our recent work, we demonstrated that Selftok can achieve near-perfect object motion generation in a toy visual environment [41].

# E    Broader Impacts

**Ethical Impacts.** Our work does not raise any ethical concerns. The research does not involve subjective assessments or the use of private data. Only publicly available datasets and models are utilized for experimentation.

**Expected Societal Implications.** Our work proposes an effective method to apply reinforcement learning in visual generation. A major societal concern with this method lies in its potential for misuse. For example, some malicious individuals may exploit our method to train model to generate violent or pornographic images. To counteract such threats, it is crucial to develop strong ethical standards and stricter regulation.

