# OpenReview forum: "Selftok-Zero: Reinforcement Learning for Visual Generation via Discrete and Autoregressive Visual Tokens"
_NeurIPS.cc/2025/Conference — NeurIPS 2025 poster_

### Official Review · Reviewer_Bvhq · 2025-06-28

**Clarity:** 1
**Significance:** 3
**Originality:** 3
**Rating:** 4
**Confidence:** 2

**Summary:**

The paper proposes an autoregressive tokenization formulation for images that relies on the reverse diffusion process. It also proposes a visual RL based generation method based on its tokenization formulation. Through experiments, it shows that the proposed method achieves improved text-to-image generation capabilities and the tokenization formulation is effective for visual RL, enabling them to apply RL-based post-training for autoregressive models.

**Questions:**

- Why is a raster-scan based solution problematic for the policy improvement optimality in RL?
- What is the key reason behind Selftok-Zero's results?

**Ethical Concerns:**

["NO or VERY MINOR ethics concerns only"]

**Final Justification:**

I have decided to keep the my original score because the writing was hard to follow.

**Quality:**

2

**Strengths And Weaknesses:**

First, I would like to point out that after multiple attempts the paper felt very foreign to me, and I am aware it could simply be that I am not familiar with some of the related works. I am open to discussing with ACs on outcomes based on my review.

One strength of the paper that stands out is that learning AR tokenization with reverse diffusion _makes sense_ and the authors showcase this idea backed by good results. The writing felt very _formal_, but for the right people, this could be appreciable. From my perspective, I would expect a clearer explanation of the two motivation put in the introduction. From its current portrayal, the motivation did not make sense to me. Many important parts are in appendix, e.g., the authors mention in line 92 "To make this tractable, we derive in Appendix a
correspondence between AR structure and reverse diffusion process by writing them in a recursive form", for the discussion of building a tokenizer, I would expect a clear understanding of the method without going to the appendix.

---

> ### Author Rebuttal · Authors · 2025-07-30
>
> Thank you very much for your constructive questions and valuable suggestions. We are delighted to discuss our work with you, and our responses are as follows:
>
> ## W1：Clarification on tokenizer and motivation
>
> We are sorry that due to page limitations, we had to place some content related to the tokenizer in appendix. Thank you for your suggestion; we will revise the paper to include more details about the tokenizer in the main text to make it easier for readers to understand.
>
> The main point of the paper is to demonstrate that autoregressive (AR) tokens is the key for reinforcement learning (RL) in visual generation. In the introduction, we explain why RL is less effective for diffusion models and AR models based on spatial tokens. To address this gap, we reveal that the reverse diffusion process is inherently AR through a recursive decomposition. This allows us to learn AR tokens that support effective visual RL.
>
>
> ## Q1：Why is a raster-scan based solution problematic for the policy improvement optimality in RL?
>
> As shown in **Figure 1** of our paper, the raster-scan based tokens do not follow the AR causal dependencies, as they are essentially inter-dependent, $e.g.$, knowing one image patch will leak information of other patches. We prove that this violates Bellman equation in **Section 2.2**.
>
> ## Q2：What is the key reason behind Selftok-Zero's results?
>
>
> As discussed in **W1**, our work aims to show that AR tokens enable effective RL-based post-training for visual generation. We constructed AR tokens as visual representations based on the autoregressive diffusion process, aligning with the policy improvement theorem. Experiments on GenEval and DPG benchmarks show its effectiveness: only AR tokens meet the conditions for the existence of an optimal solution, while other non-AR tokens yield weaker RL results—highlighting our work's efficacy.

---

### Official Review · Reviewer_EyKQ · 2025-06-28

**Clarity:** 2
**Significance:** 4
**Originality:** 4
**Rating:** 5
**Confidence:** 4

**Summary:**

The paper proposes to use Selftok (Self-consistency Tokenizer), which represents each image as a sequential 1D stream of discrete, autoregressive tokens. Then, the authors train a pure AR VLM and claim that a simple policy gradient algorithm applied to Selftok tokens significantly boosts model performance.

**Questions:**

1. I think the description in Section 2.1 lines 92–96 is too brief, which makes it difficult for a broader audience to understand. I believe that incorporating the analogy and explanation from Appendix lines 67–76 into the main paper would significantly improve comprehension.

2. Meanwhile, Section 2.2 might be overly detailed. For example, lines 135–141 do not present new content introduced by this paper, but rather a classic derivation in reinforcement learning.

3. In Appendix A line 87, there is a minor typo.

**Ethical Concerns:**

["NO or VERY MINOR ethics concerns only"]

**Final Justification:**

Overall, I believe this is a technically solid paper, with high impact on visual generation, with novel method, excellent evaluation and reliable reproducibility.

In the rebuttal, the authors resolved all my concerns, and therefore I will maintain my score of 5.

**Limitations:**

Yes.

**Quality:**

4

**Strengths And Weaknesses:**

Overall, I believe this is a technically solid paper, with high impact on visual generation, with novel method, excellent evaluation and reliable reproducibility.

However, there is also a weakness. Currently, the design of the Selftok tokenizer includes some interesting analogies, such as those in equations (A3) and (A4) in Appendix A. If the paper could provide rigorous theoretical justification or explanation for the validity of these analogies, it would make the work even more compelling.

---

> ### Author Rebuttal · Authors · 2025-07-30
>
> Thank you very much for your valuable suggestions and positive feedback. We will revise our paper in accordance with your recommendations.
>
> ## W1：Supplementary Content for Tokenizer
>
> Thank you for pointing that out! For the theoretical justification similar to our work, please refer to [1]. The theorem in this paper proves that the loss of image attributes increases as more noise is added. This thus enables adding tokens in a reverse-autoregressive manner to make up for the loss of attributes.
>
> We apologize for the insufficient coverage of content related to the tokenizer. Our work aims to emphasize that AR tokens can still support effective RL-based post-training for visual generation, so we condensed the content regarding the tokenizer. Thank you again sincerely for your suggestions. We will clarify this part in the revised version.
>
> [1] Yue Z, Wang J, Sun Q, et al. Exploring diffusion time-steps for unsupervised representation learning[J]. arXiv preprint arXiv:2401.11430, 2024.
>
> ## Q1：Supplementary Content for Tokenizer
>
> We sincerely apologize for the insufficient coverage of content related to the tokenizer. We will add more details about the tokenizer in the main text to make this section more comprehensive. Thank you again for your suggestion.
>
> ## Q2：RL-related Content
>
> We sincerely apologize for the redundancy in the proof of RL theory. We will correct it in the revised version. Thank you very much for your suggestion.
>
> ## Q3：Typo
>
> Thank you for pointing out the typo, we will correct it in the revised version.

---

> ### Comment · Reviewer_EyKQ · 2025-08-01
>
> Overall, I believe this is a technically solid paper, with high impact on visual generation, with novel method, excellent evaluation and reliable reproducibility.
>
> In the rebuttal, the authors resolved all my concerns, and therefore I will maintain my score of 5.

---

> > ### Author Response · Authors · 2025-08-04
> >
> > We sincerely appreciate your recognition of our work. And we will make sure to provide more detailed and precise descriptions in the revised version, so readers can better understand the implementation and working principles of Selftok-Zero.

---

### Official Review · Reviewer_c2No · 2025-07-02

**Clarity:** 3
**Significance:** 3
**Originality:** 3
**Rating:** 5
**Confidence:** 4

**Summary:**

This paper identifies and addresses a fundamental challenge in applying reinforcement learning (RL) to visual generation. The authors argue that existing paradigms are ill-suited for RL: diffusion models suffer from intractable policy trajectories, while standard autoregressive (AR) models using spatial tokens violate the Bellman equation, undermining the optimality of policy improvement. To resolve this, the paper proposes leveraging "Selftok," a tokenizer that represents images as truly autoregressive 1D token sequences. By training a Vision-Language Model on these tokens and then applying a policy gradient algorithm with a hybrid (program- and QA-based) reward function, the resulting model, "Selftok-Zero," achieves state-of-the-art performance on text-to-image generation benchmarks without requiring paired data during the RL phase.

**Questions:**

1.  Could you provide a more intuitive explanation of why Selftok's diffusion-based encoding ensures a stricter AR property compared to a simple raster-scan ordering of spatial patches? What is the key mechanism that prevents the "collider effect" mentioned in the introduction?

2.  The reward function is a critical component. Did you perform any ablations on the composition of the reward model? For instance, how does the model perform on a complex benchmark like DPG-Bench if trained only with the program-based reward, or vice-versa?

3.  Training large models with RL can often be unstable. Did you encounter any significant stability challenges during the policy gradient optimization? How critical was the KL-divergence term (`DKL(π||πold)`) in Eq. 10 for achieving stable convergence?

**Ethical Concerns:**

["NO or VERY MINOR ethics concerns only"]

**Limitations:**

Yes.

**Paper Formatting Concerns:**

No.

**Quality:**

3

**Strengths And Weaknesses:**

Strength

1.  The core strength of this paper is its clear and insightful diagnosis of why conventional visual tokens are problematic for RL. The argument that spatial tokens induce anti-causal dependencies that violate the policy improvement theorem (Section 2.2) is a fundamental contribution and provides a strong theoretical motivation for the proposed approach.

2.  This work is among the first to demonstrate that RL can be a highly effective post-training step for AR visual generative models. The "Selftok-Zero" concept—achieving significant performance gains purely through RL without supervised fine-tuning on new pairs—is powerful and opens a promising direction for aligning generative models with complex human preferences.

Weakness

1.  The success of the method is heavily dependent on the Selftok tokenizer, which is a complex system introduced in prior work. The main paper provides limited intuition on *how* Selftok's diffusion-based process creates truly autoregressive tokens, making the core component feel somewhat like a black box.

2.  The quality of Selftok-Zero is ultimately capped by the capabilities of the external models used for reward (e.g., InternVL, MM-Detection). Any biases, errors, or limitations in these reward models will be directly inherited and amplified by the policy. This is a general limitation of RLHF-style methods but remains a key consideration.

3. The current model generates images at a 256x256 resolution, which is low by modern standards.

---

> ### Author Rebuttal · Authors · 2025-07-30
>
> Thank you very much for your valuable comments and insightful questions. Our responses are as follows:
>
> ## W1：Supplementary Content for Tokenizer
>
> We apologize for the insufficiency of content related to the tokenizer; our work focuses on emphasizing that autoregressive (AR) tokens is the key for reinforcement learning (RL) in visual generation. Thank you again sincerely for your suggestion—we will include more details about the tokenizer in the main text.
>
> ## W2：Reward Model Design and Bias
>
> Great point! We agree that this limitation has potential influence on the policy. And we apologize that in this work, we did not explicitly address those biases, as our focus is to demonstrate that AR tokens enable effective visual RL, whereas spatial tokens are less effective in this regard. And it is worth noting that even with this potential deficiency, we still SOTA. This highlights the potential of visual RL with Selftok tokens, and we will improve the RL pipeline in our future work. This issue may need to be addressed in future work through better reward models and large-scale RL.
>
> ## W3：Image Resolution
>
> Thanks for pointing that, actually our tokenizer can be extended to 512x512, as shown in **Figure C10** in **Appendix**. We also plan to extend it to higher resolutions in future work. Furthermore, achieving competitive results at 256x256 resolution is more challenging, as generating image details is more difficult; this further illustrates the effectiveness of our method.
>
> ## Q1：Explanation of Selftok
>
> We would love to provide a causal graph that makes the explanation easier, but couldn't do so due to rebuttal policies. We will try our best to explain intuitively to you. Recall that the collider effect arises as spatial tokens are encoded by conditioning on the original image $x_1$. In our work, the key mechanism to circumvent this problem is to introduce a random noise $x_0$ independent from $x_1$. From this $x_0$, we learn tokens that gradually guide the reverse diffusion process from $x_0$ to $x_1$. Specifically, each token is learned to enable an additional step towards $x_1$, from a position $x_t$ that is dictated by **previous tokens**. This naturally leads to causal dependencies among all tokens.
>
>
> ## Q2：Reward Ablation
>
> This issue is indeed a critical component. We will show that models trained solely with program-based rewards do have improvements on DPG benchmark, with the specific results as follows:
>
> |     Model     | DPG Score |
> | :-----------: | :-------: |
> |  Selftok-SFT  |   81.80   |
> | Selftok-P$^*$ |   82.43   |
> | Selftok-Zero  |   85.57   |
>
> $*$ represents model trained only with the program-based reward
>
> We will also continue to explore more settings related to rewards in future work. Thanks again for your valuable question.
>
> ## Q3：Training Stability
>
> Nice question! We have indeed encountered challenges that significantly impact the stability of RL training. Specifically, we found that training becomes highly unstable when the learning rate is high; therefore, we set the learning rate to $1.5e−6$. Furthermore, training also becomes unstable when there is a mismatch between the model's sampling and update frequencies, so we maintain consistency between these frequencies to avoid this issue.
>
> We also explored the effects of three different KL divergence coefficients, with the results shown as follows:
>
> | KL coefficient | Geneval Score |
> | :------------: | :-----------: |
> |       0        |       \       |
> |      0.05      |      92       |
> |      0.1       |      87       |
>
> We found that when the KL coefficient was 0, training became very unstable. We tried our best to set hyperparameters, but we were unable to stabilize the training. When the KL coefficient was 0.1, the model's convergence speed slowed down, and it was unable to exceed the GenEval score when the KL coefficient was 0.05.

---

### Official Review · Reviewer_EWpd · 2025-07-02

**Clarity:** 2
**Significance:** 3
**Originality:** 3
**Rating:** 4
**Confidence:** 4

**Summary:**

This paper introduces Selftok-Zero, an autoregressive (AR) visual generative model trained with reinforcement learning (RL)), built upon Selftok’s AR visual token representation. Unlike prior models based on spatial or unstructured token sequences, Selftok-Zero leverages the AR dependency among Selftok tokens to enable stable and theoretically grounded policy improvement under RL.

**Questions:**

See weaknesses.

**Ethical Concerns:**

["NO or VERY MINOR ethics concerns only"]

**Final Justification:**

Based on the discussion with the authors, most of my concerns are resolved, except for the absence of a fair comparison with online DPO algorithms. I will raise my score, but I still strongly recommend the authors to provide such fair comparison in later versions.

**Limitations:**

yes

**Paper Formatting Concerns:**

None.

**Quality:**

3

**Strengths And Weaknesses:**

**Strength**
- The proposed Selftok-Zero method addresses the violation of the Bellman equation in the traditional AR tokenizers, enabling stable and theoretically grounded policy improvement under RL.
- The Selftok-Zero model performs well on image generation benchmarks like GenEval and DPG-Bench.

**Weakness**：
- Robustness of the Tokenization Scheme. The core of Selftok lies in establishing a correspondence between the Autoregressive (AR) structure and the diffusion process. To what extent does this correspondence depend on the choice of the diffusion process q(xt|x1) and the token schedule k(t)? Could different scheduling strategies (e.g., linear, cosine) or noise types significantly impact the quality and structure of the final AR tokens?
- Semantic Interpretability of Tokens. While traditional spatial tokens may be causally flawed, their positional information makes them somewhat interpretable. For the AR token sequence generated by Selftok, do individual tokens or consecutive token segments correspond to specific, interpretable semantic concepts in the image (e.g., object contours, textures, or parts)? Could the authors provide a visual analysis to reveal the correspondence between these AR tokens and the image content?
- Reward Model Design and Bias. The two proposed reward models (Program-based and QA-based) are clever, but they may introduce jejich own biases. For instance, the Program-based Reward depends on the performance of a specific detector; if the detector performs poorly on certain objects, the RL process could be misguided. Similarly, the QA-based Reward relies on the inherent biases and knowledge limitations of the "judge" VLM (e.g., GPT-4o). How do the authors assess and mitigate the potential biases introduced by these reward models themselves?
- Direct Comparison with Competing Works. A core argument of the paper is that methods like Diffusion-DPO are inefficient due to the mismatch between the behavior and target policies. Could the authors design an experiment to directly compare the performance of Selftok+RL against Diffusion-DPO under identical settings?
- Quantitative Evaluation of Encoder Quality and Efficiency. To provide a comprehensive evaluation of the proposed tokenizer, could the authors report on several key quantitative metrics(rFID, the time or FLOPs required to encode a single image)?

---

> ### Author Rebuttal · Authors · 2025-07-30
>
> Thank you very much for your constructive comments and invaluable evaluation. We are pleased to see that you have raised very critical questions regarding our work, and our responses are as follows:
>
> ## W1：Robustness of the Tokenization Scheme.
>
> Nice observation! The choice of the diffusion process $q(x_t|x_1)$ and the token schedule $k(t)$ should ensure that all decompositions $P(V_K)=P(V_{{\lt}i})P(V_{{\geq} i}|V_{{\lt}i})$ are accounted for in training. For example, consider a failure case where $x_t=x_0$ for any $t\neq1$, or a token schedule such that all tokens are allocated to $V_{{\ge}1}$, $i.e.$, $k(t)=1$, $∀𝑡 ∈ [0, 1)$. This corresponds to a trivial decomposition$P(V_{{<}1} = [\space])·P(V_K|V_{{<}1} = [\space])$, $V_K ⇔ x_0 ⇝ x_1$, and $[\space] ⇔ x_0$ where we always input the full $V_K$ to the decoder. So $V_K$ loses all the AR property.
>
> For the forward process, we adopt rectified flow ($i.e.$, $x_t=tx_1+(1-t)x_0$), as we initialize our decoder from SD3. We are happy to try other scheduling strategies and noise types as future work.
>
> For token schedule, as we uniformly sample $𝑡 ∈ [0, 1]$ in training, the best token schedule should be a uniform assignment $𝑘^∗(𝑡) = ⌈𝑡 × 𝐾⌉ + 1$ to ensure that every decomposition is equally respected in the training objective. We perform ablations in **Token schedule** of **Appendix C**. Interestingly, we observe a slightly better reconstruction quality by designing a schedule $𝑘(𝑡)$ that allocates fewer tokens to smaller $𝑡$, $i.e.$, $𝑘(𝑡) < 𝑘^∗(𝑡) \space for \space 𝑡< 0.5$. This aligns with the well-known trait of diffusion models: the early path $x_0 ⇝ x_𝑡$ for a small $𝑡$ has minimal impact on the reconstruction $x_𝑡 ⇝ x_1$, which can be omitted.
> ## W2：Semantic Interpretability of Tokens
>
> Yes, we find that tokens corresponding to smaller time-steps tend to capture the overall background, color tone or composition of the image, those at middle ones tend to capture object shapes and those at larger ones tend to capture fine-grained details and textures. This is because the diffusion process itself is tightly linked with visual semantics [1,2,3], and Selftok simply encode the process as tokens.
>
> We would love to show you some visualizations that demonstrate the interpretability of our tokens, but we are unable to do so due to the constraints of the conference policy. We will follow your suggestion to include those figures in the revision. Meanwhile, you can check similar results in **Figure 1** of [1].
>
> [1] Yue Z, Wang J, Sun Q, et al. Exploring diffusion time-steps for unsupervised representation learning[J]. arXiv preprint arXiv:2401.11430, 2024.
>
> [2] Preechakul K, Chatthee N, Wizadwongsa S, et al. Diffusion autoencoders: Toward a meaningful and decodable representation[C]. Proceedings of the IEEE/CVF conference on computer vision and pattern recognition. 2022: 10619-10629.
>
> [3] Zhang Z, Zhao Z, Lin Z. Unsupervised representation learning from pre-trained diffusion probabilistic models[J]. Advances in neural information processing systems, 2022, 35: 22117-22130.
>
> ## W3：Reward Model Design and Bias
>
> Great point! We agree that this could introduce potential biases. In this work, we did not explicitly address those biases, as our focus is to demonstrate that AR tokens enable effective visual RL, whereas spatial tokens are less effective in this regard. And it is worth noting that even with this potential deficiency, we still achieve state-of-the-art performance. This highlights the potential of visual RL with Selftok tokens, and we will improve the RL pipeline in our future work.
>
> ## W4：Direct Comparison with Competing Works
>
> We fine-tuned a diffusion model (SDXL) with Diffusion-DPO using the same reward model on the GenEval benchmark. To ensure the identical settings, we train SDXL with the same amount of data, and we used prompts and images to construct positive and negative pairs, whereas Selftok-Zero only use prompts. The results are as follows:
>
> |       Methods        | GenEval Score |
> | :------------------: | :-----------: |
> |         SDXL         |     53.8      |
> | SDXL + Diffusion-DPO |  56.3(+2.5)   |
> |     Selftok-SFT      |      74       |
> |     Selftok-Zero     |    92(+18)    |
>
> The table above shows that Diffusion-DPO does not exhibit a significant improvement(+2.5) on the GenEval benchmark, while our method leads to a +18 increase in overall score, which further demonstrates the effectiveness of our method.
>
> ## W5：Quantitative Evaluation of Encoder Quality and Efficiency
>
> The comparisons with other works, as well as the results of quantitative metrics related to the Tokenizer, are as follows:
>
> |     Tokenizer      | Type | #Token |   #Code   |  rFID↓   |   PSNR↑   |   SSIM↑   |  LPIPS↓   |
> | :----------------: | :--: | :----: | :-------: | :------: | :-------: | :-------: | :-------: |
> |      LlamaGen      |  2D  |  1024  | $2^{14}$  |   0.59   |   24.44   |   0.768   |   0.064   |
> |       Cosmos       |  2D  |  1024  | $≈2^{16}$ |   0.87   |   24.82   |   0.763   |   0.070   |
> |        VAR         |  2D  |  680   | $2^{12}$  |   0.99   |   22.12   |   0.624   |   0.109   |
> |    TiTok-S-128     |  1D  |  128   | $2^{12}$  |   1.71   |   17.52   |   0.437   |   0.210   |
> |      FlexTok       |  1D  |  256   |   64000   |   1.45   |   18.53   |   0.465   |   0.222   |
> |     FlowMo-Hi      |  1D  |  1024  | $2^{14}$  |   0.56   |   24.93   |   0.785   |   0.073   |
> | **Selftok (Ours)** |  1D  |  1024  | $2^{15}$  | **0.54** | **26.30** | **0.805** | **0.063** |
>
> | Time(s) | TFLOPs |
> | :-----: | :----: |
> |  0.86   |  2.59  |
>
>
> From the table above, it can be observed that our tokenizer has achieved the best performance to date, reaching optimal levels in metrics such as rFID (0.54), PSNR (26.30), SSIM (0.805), and LPIPS (0.063). Additionally, it boasts high computational efficiency, requiring only 0.86 seconds to encode and decode an image.

---

> > ### Comment · Reviewer_EWpd · 2025-08-08
> >
> > I would like to thank the authors sincerely for the detailed response. Let's discuss these points one by one.
> >
> > W1:Thank you very much for your response. It demonstrates that in your experiments, you directly chose a uniform (0, 1) schedule. In the SD3 paper, there is a detailed analysis and experiments regarding schedule, particularly on whether to add more steps in the early denoising stage and obtain a precise flow shift the schedule. Since de-tokenization is a crucial component of the paper, I believe the analysis in this section is not sufficiently thorough. Currently, the ablation study does not provide a comprehensive discussion on this issue, which makes it difficult to fully validate the conclusions.
> >
> > W2&W3: problem solved.
> >
> > W4: Thank you for your reply. However, I believe that directly comparing offline Diffusion-DPO with online selftok-RL is unfair. The selftok-RL approach benefits from using metric-based QA & program rewards in an online manner, which can naturally lead to better results. Adding experimental results for online DPO to further demonstrate the superiority of your proposed method can enhance this paper.
> >
> > W5: problem solved.

---

> > > ### Author Response · Authors · 2025-08-08
> > >
> > > Thank you sincerely for your discussion with us. Our responses are as follows.
> > >
> > > ## Response to W1
> > >
> > > We respectfully disagree with your opinion regarding the schedule. First, we have not fully understanded the specific meaning of "schedule" in this context. If it refers to the noise schedule, since our work builds upon SD3, we directly adopted the optimal schedule identified by SD3 for subsequent research. If it refers to the token schedule, we have provided relevant ablation studies in **Appendix C**. Furthermore, the core focus of our work is to demonstrate that AR tokens can support effective RL-based post-training for visual generation, so we are unable to allocate extensive space to discussions of the schedule.
> > >
> > > ## Response to W4
> > >
> > > We apologize that we have not fully understood the specific meaning of online DPO. Could it refer to having the model construct preference data through sampling before each update? However, the online format makes it difficult to ensure the stable construction of preference data for continuous training, and constructing data in an online manner is also computationally expensive. Furthermore, even if we were able to find a way to address the aforementioned issues, we would not have sufficient time to conduct relevant experiments before the deadline of the discussion phase. Thus, it is not practical for us.

---

> > > > ### Comment · Reviewer_EWpd · 2025-08-08
> > > >
> > > > W1：Thank you very much for your response. We appreciate your clarification and acknowledge that your work includes meaningful exploration of token schedule designs. However, we suggest refining the token schedule figure and the corresponding descriptions in the manuscript to prevent potential misunderstandings.
> > > >
> > > > W4:Your understanding is correct: “online” refers to constructing preference data by sampling from the model before each update. Given that your RL approach also constructs new training data with reward signals via sampling before each update, comparing it solely against DPO methods that use only offline preference datasets is not a fair comparison. Furthermore, it should be noted that your method could be readily adapted to use the pairwise ranking [1], rather than relying on scalar reward signals. This would make the comparison with DPO both more direct and more meaningful. I encourage the authors to clarify this point.
> > > >
> > > > [1] Direct Language Model Alignment from Online AI Feedback

---

> > > > > ### Author Response · Authors · 2025-08-09
> > > > >
> > > > > ## Response to W1
> > > > >
> > > > > Thank you very much for your suggestion. We will revise the relevant content in the revised version to make the content on the token schedule more accessible and easier for readers to understand.
> > > > >
> > > > > ## Response to W4
> > > > >
> > > > > Thank you very much for your explanation. The main focus of our work is to demonstrate that AR tokens can support effective RL-based post-training for visual generation. To validate the effectiveness of our method, we have also conducted experiments on Janus Pro 7B under identical online settings, with the experimental results as follows:
> > > > >
> > > > > |       Model       | GenEval |  DPG-Bench   |
> > > > > | :---------------: | :-----: | :----------: |
> > > > > |   Janus-Pro-7B    |   79    |    83.48     |
> > > > > | Janus-Pro-7B-Zero | 85(+6)  | 84.49(+1.01) |
> > > > > |    Selftok-SFT    |   74    |    81.80     |
> > > > > |   Selftok-Zero    | 92(+18) | 85.57(+3.77) |
> > > > >
> > > > > The results are also presented in the paper's **Table 1** and **Table 2**. The experiments conducted on Janus Pro are also online; however, since Janus Pro's tokens lack the AR property, the performance of RL training is less pronounced than ours. Comparisons with offline methods such as Diffusion-DPO further validate the effectiveness of our method. Additionally, we sincerely appreciate your suggestion of using pairwise ranking to improve our method, and we will explore this in future work.

---

### Note · Authors · 2025-08-15

Dear AC and Reviewers,

We sincerely thank the AC and all reviewers for your dedicated work, constructive comments, and recognition of our work. We are pleased many reviewers acknowledged our work’s key strengths:
- **Solid theoretical foundation**: Our work details why AR visual tokens satisfy effective RL conditions and highlights that spatial tokens violate causal dependencies, failing to support efficient RL.(EWpd: "**stable and theoretically grounded**", c2No: "**insightful**")
- **Innovative methodology**: Our work leverages reverse diffusion’s AR nature to construct AR tokens, achieving significant improvements without supervision.(c2No: "**among the first**")
- **Significant performance improvements**: Our work achieves SOTA on GenEval and DPG-Bench, with all reviewers recognizing our experimental results.

***Summary of Main Points Raised by Reviewers***
- **Motivation**: Questions on Selftok-Zero’s motivation(Bvhq)
- **Tokenizer details**: Questions on token schedule, semantic interpretability, efficiency, theories, and requests for providing content(all)
- **Reward design**: Questions on reward design/bias and requests for reward ablation studies(EWpd, c2No)
- **RL setting**: Questions on training stability and comparisons with existing methods(EWpd, c2No)
- **Manuscript revisions**: Requests to revise tokenizer and RL theorem content(all)

***Summary of Responses***
- **Motivation**: Clarified motivation and explained why spatial tokens fail the Bellman equation.
- **Tokenizer details**: Supplemented token schedule content/results; added semantic interpretability references/decoding explanations; included metrics; supplemented theoretical references.
- **Reward design**: Added reward ablation results; acknowledged exploring reward bias mitigation in future work.
- **RL setting**: Identified training stability factors; provided KL divergence ablation results; added comparisons with offline Diffusion-DPO and Janus-Pro online results.

***Revisions and Additions in the Updated Manuscript***
1. **Expanded Context and Interpretation**:
   - Supplement tokenizer theory/implementation details in main text.
   - Add semantic interpretability visual analysis.
   - Detail token schedule content/experiments.
2. **Additional Results and Metrics**:
   - Add tokenizer results/metrics.
   - Add RL ablation results and Diffusion-DPO comparisons.
3. **Simplified Content**:
   - Simplify RL theoretical derivation in Section 2.2 for readability.

---

### Decision · Program_Chairs · 2025-09-17

**Decision:**

Accept (poster)

**Comment:**

The authors introduce SelfTok-Zero, an AR model that builds on top of previous tokenizer SelfTok, enhanced with RL-based post-training. Effectiveness of the proposed method is demonstrated on GenEval and DPG-Bench.

Initially, the reviewers raised several concerns, which are briefly outlined below:

* Reviewer EWpd: Tokenization robustness, interpretability of tokens, reward model design, method comparison, tokenizer evaluation.

* Reviewer c2No: Dependency on prior work Selftok tokenizer and reward models, limited resolution generation.

* Reviewer EyKQ: Writing and method clarification.

* Reviewer Bvhq: Method's motivation.

The rebuttal and subsequent author-reviewer discussions effectively addressed most of the reviewers' concerns. After carefully considering the reviews, rebuttal, and discussion, the AC concurs with the reviewers’ assessment and thus recommends acceptance of the paper. Finally, the authors are encouraged to incorporate the rebuttal experiments into the manuscript and address the reviewers’ feedback (e.g., the writing refinement) in the final revision.